Report

EMBO
Molecular Medicine

# A dual-AAV approach restores fast exocytosis and partially rescues auditory function in deaf otoferlin knock-out mice

Hanan Al-Moyed[1,2], Andreia P Cepeda[1,2] (iD), SangYong Jung[3,4], Tobias Moser[2,3,4] (iD), Sebastian Kügler[5,*] (iD) & Ellen Reisinger[1,**] (iD)

## Abstract

Normal hearing and synaptic transmission at afferent auditory inner hair cell (IHC) synapses require otoferlin. Deafness DFNB9, caused by mutations in the *OTOF* gene encoding otoferlin, might be treated by transferring wild-type otoferlin cDNA into IHCs, which is difficult due to the large size of this transgene. In this study, we generated two adeno-associated viruses (AAVs), each containing half of the otoferlin cDNA. Co-injecting these dual-AAV2/6 half-vectors into the cochleae of 6- to 7-day-old otoferlin knock-out ($Otof^{-/-}$) mice led to the expression of full-length otoferlin in up to 50% of IHCs. In the cochlea, otoferlin was selectively expressed in auditory hair cells. Dual-AAV transduction of $Otof^{-/-}$ IHCs fully restored fast exocytosis, while otoferlin-dependent vesicle replenishment reached 35–50% of wild-type levels. The loss of 40% of synaptic ribbons in these IHCs could not be prevented, indicating a role of otoferlin in early synapse maturation. Acoustic clicks evoked auditory brainstem responses with thresholds of 40–60 dB. Therefore, we propose that gene delivery mediated by dual-AAV vectors might be suitable to treat deafness forms caused by mutations in large genes such as *OTOF*.

**Keywords** deafness; gene therapy; hearing restoration; inner ear; inner hair cell
**Subject Categories** Genetics, Gene Therapy & Genetic Disease; Neuroscience

See also: **JR Holt & GSG Geleoc** (January 2019)

## Introduction

Mutations in the *OTOF* gene cause profound congenital non-syndromic autosomal recessive hearing loss (DFNB9, Yasunaga *et al*, 1999). $Otof^{-/-}$ mice or animals with deleterious point mutations in this gene are profoundly deaf (Roux *et al*, 2006; Longo-Guess *et al*, 2007; Pangrsic *et al*, 2010; Reisinger *et al*, 2011). The *OTOF* gene encodes otoferlin, a large multi-C$_2$-domain protein predominantly expressed in inner hair cells (IHCs), the genuine sensory cells in the cochlea. IHCs transform mechanical acoustic vibrations into a neural code via synaptic transmission to auditory neurons. Studies using $Otof^{-/-}$ and *Otof* knock-in mouse models revealed that this protein plays an essential role in IHC exocytosis and vesicle replenishment, and is involved in vesicle reformation and endocytosis (Roux *et al*, 2006; Pangrsic *et al*, 2010; Duncker *et al*, 2013; Jung *et al*, 2015; Strenzke *et al*, 2016). To date, over one thousand pathogenic mutations have been identified within this gene, affecting 2.3–10% of patients from various ethnicities suffering from hereditary non-syndromic hearing loss (Rodríguez-Ballesteros *et al*, 2008; Choi *et al*, 2009; Iwasa *et al*, 2013; Van Camp & Smith). In contrast to many other deafness or Usher genes, otoferlin seems dispensable for auditory hair cell (HC) development (Roux *et al*, 2006). Given the normal inner ear morphology in these patients, a postnatal transfer of otoferlin cDNA into the inner ear is predicted to ameliorate this hearing loss. Gene therapy might outperform the otherwise necessary cochlear implantation, which transmits only part of the acoustic information. Cochlear implant users often report difficulties in speech understanding during noise and in perceiving vocal emotions, and typically cannot experience music as a person without hearing impairment (Fu *et al*, 1998; Nelson *et al*, 2003; McDermott, 2004; Luo *et al*, 2007; Most & Aviner, 2009; Oxenham & Kreft, 2014; Chatterjee *et al*, 2015; Paquette *et al*, 2018). Yet, no delivery method for large genes, like electroporation or viral transduction via adenoviruses, lentiviruses, or semliki forest viruses, transferred cDNA into HCs in the postnatal inner ear with high efficiency. Recombinant adeno-associated viruses (AAVs) are a safe and promising gene therapy tool to treat hearing impairment (reviewed in Ref. Géléoc & Holt, 2014). Prenatal injections of AAVs into the

1 Molecular Biology of Cochlear Neurotransmission Group, Department of Otorhinolaryngology, University Medical Center Göttingen, and Collaborative Research Center 889, University of Göttingen, Göttingen, Germany
2 Göttingen Graduate School for Neurosciences, Biophysics, and Molecular Biosciences, University of Göttingen, Göttingen, Germany
3 Institute for Auditory Neurosciences and InnerEarLab, University Medical Center Göttingen, Göttingen, Germany
4 Synaptic Nanophysiology Group, Max Planck Institute for Biophysical Chemistry, Göttingen, Germany
5 Center Nanoscale Microscopy and Physiology of the Brain (CNMPB), Department of Neurology, University Medical Center Göttingen, Göttingen, Germany
*Corresponding author. Tel: +49 551 39 8351; E-mail: sebastian.kuegler@med.uni-goettingen.de
**Corresponding author. Tel: +49 551 39 9688; E-mail: ellen.reisinger@med.uni-goettingen.de

developing otocyst or postnatal cochlear injections have been proven to efficiently transduce IHCs in animal models (Liu *et al*, 2005; Bedrosian *et al*, 2006). However, the limited AAV cargo capacity of approximately 4.7–5 kb presents an obstacle for the transfer of large coding sequences (CDS) such as the 6 kb-long otoferlin cDNA. Split-AAV vectors, each containing a fragment of the large transgene expression cassette, have been developed to circumvent this problem. This technique takes advantage of the intrinsic ability of the AAV genome to form tail-to-head concatemers by end-joining of its inverted terminal repeats (ITRs) (Duan *et al*, 1998). In the "trans-splicing (TS)" strategy, the ITRs are spliced out after transcription by introducing artificial splice donor (SD) and acceptor (SA) sites before and after the ITRs in the respective half-vectors. The reconstitution of the large transgene can also be mediated through homologous recombination of overlapping sequences placed at the 3′-end of the first AAV half-vector and at the 5′-end of the second in the "overlap" split-AAV strategy. The "hybrid" strategy is a combination of both previous strategies (Ghosh *et al*, 2008). The correct reassembly of the full-length expression cassette in the nuclei of target cells results in the production of full-length mRNA and protein (Yan *et al*, 2000; Duan *et al*, 2001; Chamberlain *et al*, 2016). To date, dual- and triple-AAV vectors have demonstrated efficacy in photoreceptors and muscle cells (Duan *et al*, 2001; Ghosh *et al*, 2008; Trapani *et al*, 2014; Maddalena *et al*, 2018), but have not been established for IHCs.

## Results and Discussion

In this study, we investigated whether the delivery of full-length otoferlin cDNA into IHCs via dual-AAV vectors can restore defective IHC and auditory functions in $Otof^{-/-}$ mice. We aimed for a rather late time point of treatment, since the early development of the inner ear does not seem to require otoferlin (Roux *et al*, 2006), making gene therapy of mature $Otof^{-/-}$ IHCs feasible in theory. AAVs were injected into the cochlea at postnatal day 6–7 (P6–7) because the auditory bulla structure covering the round window membrane (RWM) is still soft enough at this developmental stage to be penetrated well with an injection glass pipette. We chose AAVs with ITRs from serotype 2 and capsid proteins from serotype 6 (AAV2/6) as they can be produced with an excellent transducing-unit to vector-genome ratio (Grieger *et al*, 2016). This prevents administration of excess protein bolus into the delicate structure of the inner ear. To test whether this AAV serotype transduces IHCs efficiently, we injected single AAV2/6 viruses coding for eGFP through the RWM into the scala tympani of the left cochlea of CD1xC57BL/6N-F1 (CD1B6F1) wild-type mice. eGFP fluorescence was observed in IHCs, outer hair cells (OHCs), supporting cells, and spiral ganglion neurons (SGNs), indicating that the AAV2/6 has no specific IHC tropism and targets a variety of different cell types within the organ of Corti (Fig EV1A). 34–99% of IHCs (average = 77 ± 4%, mean ± standard error of the mean (s.e.m., $n = 7$ cochleae) exhibited eGFP fluorescence, revealing a high IHC transduction efficiency of the AAV2/6. This finding contrasts recent reports showing that AAV2/6 failed to transduce IHCs if injected at P1–2 (Shu *et al*, 2016). We assume that vector quality, titer, the injection procedure itself, and the animal age at the time of surgery are all factors influencing IHC transduction efficiency.

For gene replacement therapy in $Otof^{-/-}$ mice, we used mouse otoferlin transcript variant 4 cDNA (NM_001313767), coding for the

1977 amino acid-long protein and previously confirmed to be expressed endogenously in wild-type cochleae (Strenzke *et al*, 2016). We designed otoferlin dual-AAV-trans-splicing (dual-AAV-TS) and dual-AAV-hybrid (dual-AAV-Hyb) half-vectors, both containing the N-terminal otoferlin CDS in the 5′-AAV half-vector and the C-terminal CDS in the 3′-AAV vector (Fig EV2). Expression from the 5′AAV is driven by a human β actin promoter/CMV enhancer and additionally codes for a separately translated eGFP-fluorescent reporter to identify transduced cells in acutely isolated organs of Corti. These split-AAV vectors were co-injected through the RWM into the left cochlea of P6–7 CD1B6F1-$Otof^{-/-}$ mice. Organs of Corti from these animals were isolated at P18–30 and immunolabeled with two otoferlin antibodies, one binding within the N-terminal part of otoferlin and the other one binding after the transmembrane domain close to the C-terminus of otoferlin (Figs 1A–C and EV3, Appendix Fig S1–S3). Upon dual-AAV injection into $Otof^{-/-}$ cochleae, we found otoferlin immunofluorescence to be restricted to auditory HCs, with stronger expression in IHCs and much weaker in sparsely transduced OHCs (Appendix Fig S3), resembling otoferlin expression in wild-type animals (Roux *et al*, 2006; Beurg *et al*, 2008). eGFP fluorescence, on the contrary, was also found in other cell types (Figs 1A and EV1B, Appendix Fig S3), although the expression of both proteins is driven by the same promoter and they are translated from the same mRNA (Fig EV2). Neither eGFP nor otoferlin expression could be detected in contralateral non-injected ears (Fig 2A, Appendix Fig S1 and S2). We presume that a yet unknown mechanism such as post-transcriptional regulation or targeted protein degradation restricts the expression of otoferlin to auditory HCs. A similar finding was reported for AAV1 postnatal RWM injections (P1–3 and P10–12), where Vglut3 expression was found selectively in IHCs despite the broad cell type tropism of this AAV serotype (Akil *et al*, 2012). The restricted expression of otoferlin to HCs would be very beneficial for human gene therapy applications to avoid potential off-target effects due to expression of exogenous otoferlin in non-sensory cells in the inner ear.

The number of IHCs immunolabeled with both N-terminal and C-terminal otoferlin antibodies along the entire cochlea ranged from 12 to 51% in dual-AAV-TS (average: 30 ± 4%, $n = 10$ animals) and from 5 to 34% in dual-AAV-Hyb (average: 19 ± 3%, $n = 9$ animals) injected $Otof^{-/-}$ mice (Fig 1D). Approximately 10% of IHCs were solely labeled by the N-terminal otoferlin antibody, likely indicating no correct reassembly of the two virus half-vectors in on average one out of four transduced cells (Figs 1D and EV3). We did not find any IHC showing only a C-terminal otoferlin immunofluorescence signal, which was expected since the 3′-AAV half-vector does not contain a separate promoter. A previous study indicated that 70% of intact IHCs suffice for proper auditory function (Wang *et al*, 1997). Further studies will reveal if optimizing the virus administration procedure into the cochlea (e.g., as in ref. Yoshimura *et al*, 2018) might increase the otoferlin IHC transduction rate.

In order to examine whether the split full-length otoferlin expression cassette reassembled to produce the correct full-length mRNA in the target cells, we isolated mRNA from transduced $Otof^{-/-}$ organs of Corti (P26–29) and amplified otoferlin cDNA fragments spanning the dual-AAV split-site (Fig EV4). From dual-AAV-transduced $Otof^{-/-}$ organs of Corti and wild-type control samples, we amplified a PCR product with the size expected for full-length otoferlin cDNA (1,753 bp) that was absent in non-injected $Otof^{-/-}$

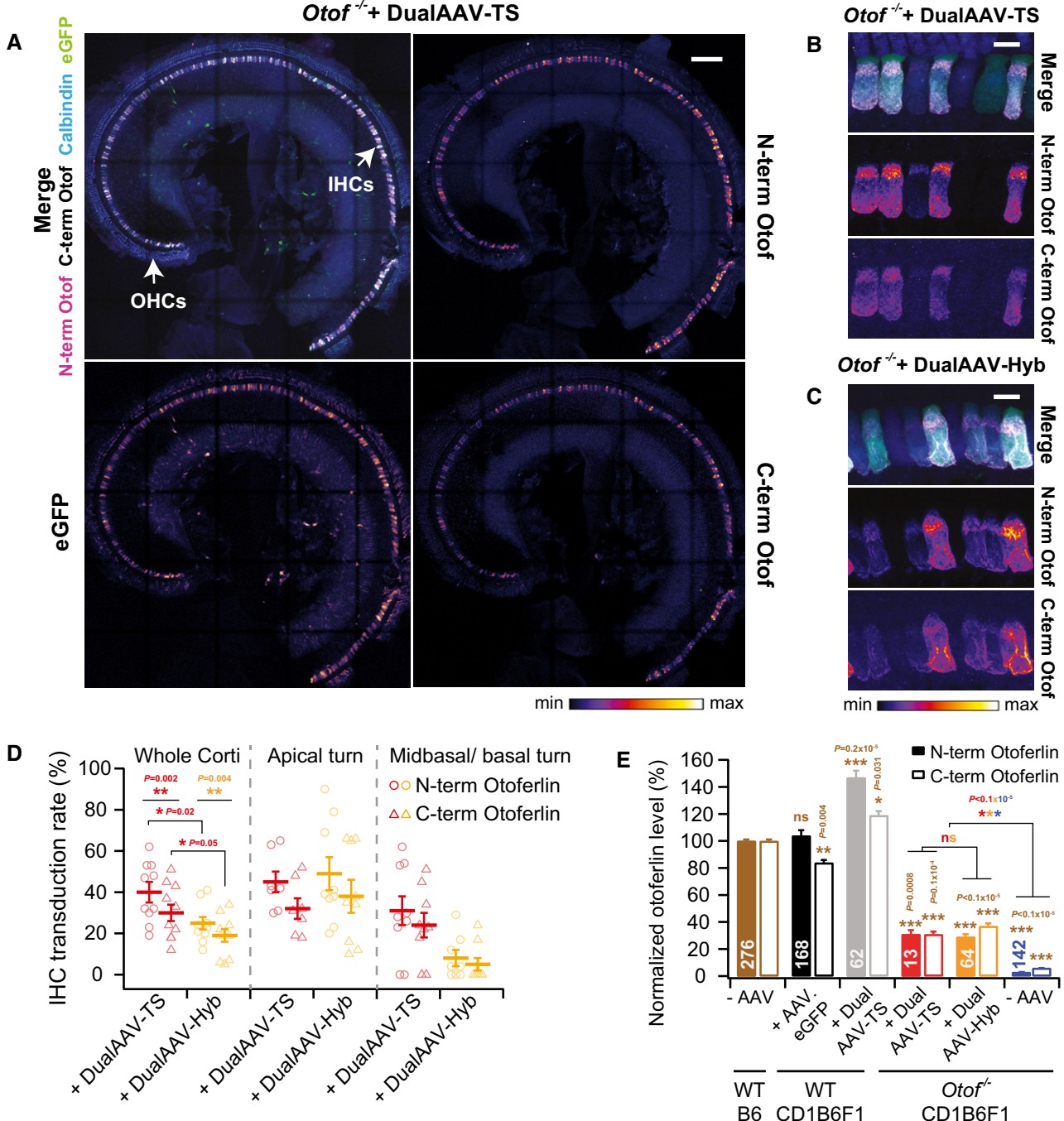

**Figure 1.   Dual-AAV-mediated otoferlin expression is restricted to auditory hair cells in the cochlea.**

A      Low magnification views of a CD1-*Otof*$^{-/-}$ organ of Corti (P23)-transduced with otoferlin dual-AAV-TS vectors. IHCs: inner hair cells, OHCs: outer hair cells.

B, C   High magnification views of CD1B6F1-*Otof*$^{-/-}$ IHCs transduced with otoferlin dual-AAV-TS (P26) (B) and dual-AAV-Hyb (P26) (C) vectors. Individual eGFP and otoferlin immunostainings are depicted as color lookup tables in (A-C) with warmer colors representing higher pixel intensities. See Fig EV3 for comparison to wild-type IHCs.

D      Percentage of N- and C-terminal otoferlin labeled IHCs in dual-AAV-TS (*n* = 10 mice)- and dual-AAV-Hyb (*n* = 9 mice)-injected CD1B6F1-*Otof*$^{-/-}$ mice (P18–30).

E      Average N-terminal and C-terminal otoferlin immunofluorescence levels in dual-AAV-transduced *Otof*$^{-/-}$ and wild-type IHCs (P23–30). Otoferlin levels were normalized to immunofluorescence levels in non-transduced B6 wild-type IHCs for each antibody separately.

Data information: In (A–C), maximum intensity projections of confocal optical sections. Scale bars: 100 μm (A), 10 μm (B, C). In (D), individual animals are depicted with open symbols. In (E), the number of quantified IHCs is indicated inside the bars. In (D, E), data are displayed as mean ± s.e.m., ns *P* > 0.05; *$P$ ≤ 0.05; **$P$ ≤ 0.01; ***$P$ ≤ 0.001, [Wilcoxon matched-pair signed rank test (D), unpaired t-test with Welch's correction (D), and Kruskal–Wallis test followed by Dunn's multiple comparison test (E)].

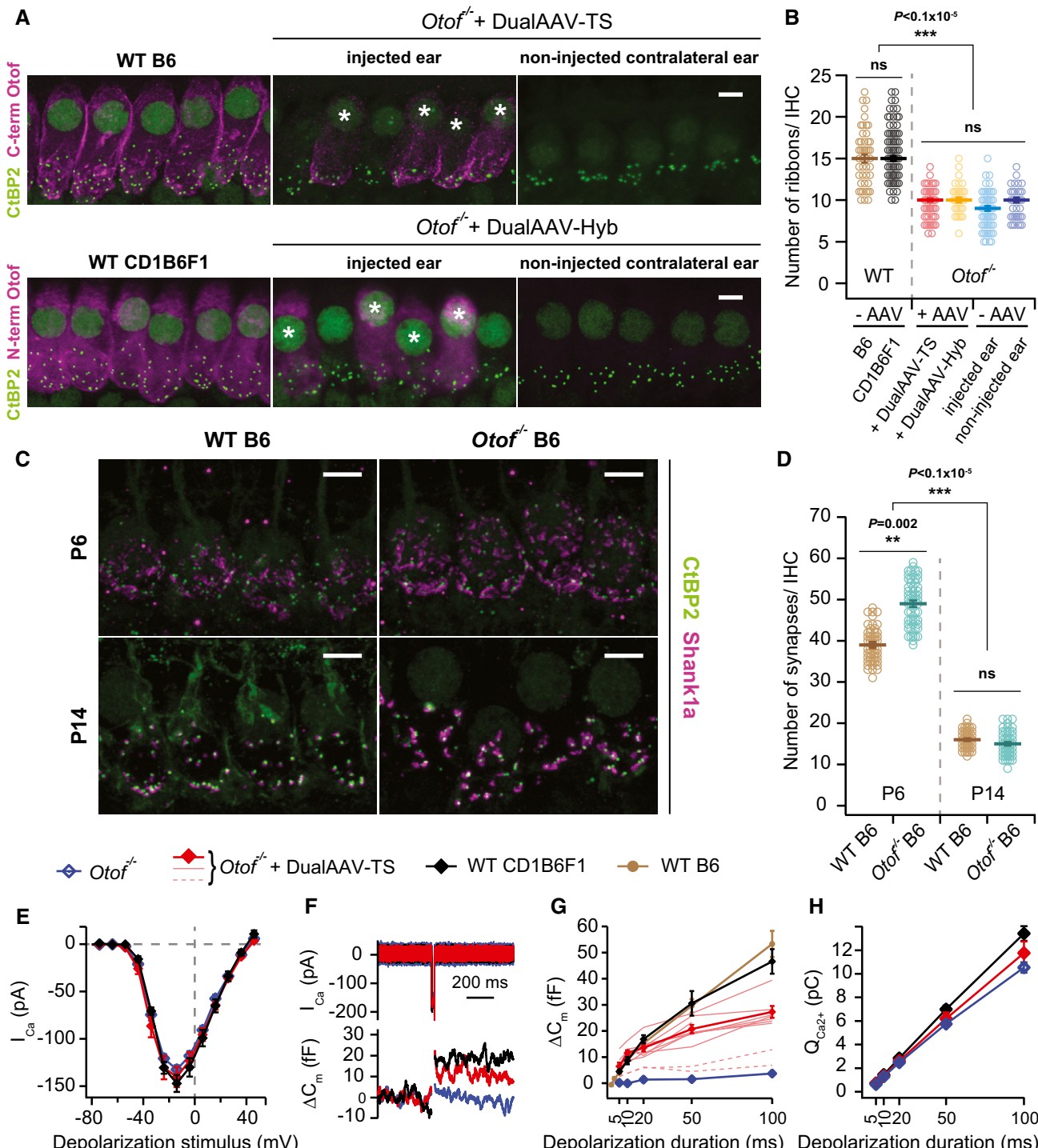

**Figure 2.**

organ of Corti samples (Fig EV4A and B). Amplicons from wild-type, control $Otof^{-/-}$, and dual-AAV-TS-transduced $Otof^{-/-}$ samples were subcloned and representative clones were subjected to Sanger sequencing ($n = 2$ clones/4 clones/5 clones, respectively; Fig EV4B and C). This confirmed the correct reconstitution of the full-length transgene from the two otoferlin AAV half-vectors and the presence of an artificially introduced AccIII restriction site found only in the dual-AAV-transduced $Otof^{-/-}$ samples (Fig EV4C). In cDNA

samples of dual-AAV-injected and non-treated $Otof^{-/-}$ organs of Corti, we amplified three otoferlin cDNA fragments of 1,379, 1,480, and 1,679 bp, all lacking exons 14 and 15 (Fig EV4B and C). The larger amplicons originate from incomplete splicing of the mutant mRNA (Fig EV4C). These splice variants might be translated into shorter fragments, the presence of which we assessed by Western blot (Fig EV4D). In wild-type organs of Corti, we detected two specific bands of ~210–230 kDa, likely corresponding to full-length

**Figure 2.  Otoferlin dual-AAV injection at P6–7 partially restores synaptic function in $Otof^{-/-}$ IHCs.**

A    High magnification views of IHCs immunolabeled for otoferlin and synaptic ribbons (CtBP2) from wild-type (B6: P27, CD1B6F1: P29), dual-AAV-injected CD1B6F1-$Otof^{-/-}$ (dualAAV-TS: P26, dualAAV-Hyb: P28), and their contralateral non-injected ears. (*) Transduced cells. Maximum intensity projections of optical confocal sections. Scale bars: 5 μm.

B    Synaptic ribbon numbers quantified from IHCs in apical cochlear turns of wild-type (B6: $n$ = 48 IHCs, CD1B6F1: $n$ = 108 IHCs), transduced $Otof^{-/-}$ (dualAAV-TS: $n$ = 59 IHCs, dualAAV-Hyb: $n$ = 37 IHCs), and non-transduced $Otof^{-/-}$ IHCs from injected (-AAV-injected ear, $n$ = 65 IHCs) and contralateral non-injected (-AAV non-injected ear, $n$ = 46 IHCs) ears (P25–29).

C    IHC synapses labeled with CtBP2 and the postsynaptic marker Shank1a in B6 wild-type and $Otof^{-/-}$ P6 and P14 organs of Corti. Maximum intensity projections of optical confocal sections. Scale bars: 5 μm.

D    Synapse numbers quantified from IHCs in apical cochlear turns (C) of B6 wild-type (P6: $n$ = 53 IHCs; P14: $n$ = 73 IHCs) and B6-$Otof^{-/-}$ (P6: $n$ = 62 IHCs; P14: $n$ = 65 IHCs) mice at two different developmental stages (P6 and P14).

E    $Ca^{2+}$-current–voltage relationship of control CD1B6F1 wild-type ($n$ = 6 IHCs), dual-AAV-TS-transduced ($n$ = 8 IHCs), and non-transduced CD1B6F1-$Otof^{-/-}$ ($n$ = 10 IHCs) IHCs (P14–18).

F    Representative $Ca^{2+}$-currents ($I_{ca}$) and IHC plasma membrane capacitance increments ($\Delta C_m$) of a wild-type control, transduced, and non-transduced $Otof^{-/-}$ IHC in response to a 20 ms depolarization pulse at maximum $Ca^{2+}$-current potentials (typically −14 mV).

G, H    Average exocytosis level measured as $\Delta C_m$ (G) and corresponding $Ca^{2+}$-current integrals ($Q_{Ca2+}$) (H) in wild-type [CD1B6F1: $n$ = 6 IHCs; B6: $n$ = 11 IHCs (B6 data replotted from Strenzke et al, 2016)], dual-AAV-TS-transduced $Otof^{-/-}$ ($n$ = 8 IHCs), and non-transduced ($n$ = 11 IHCs) $Otof^{-/-}$ IHCs. Individual dual-AAV-TS transduced $Otof^{-/-}$ IHCs expressing eGFP, that had exocytosis (thinner red lines) and almost no exocytosis (broken red lines; not included into the average), are depicted.

Data information: In (B, D), individual animals are depicted with open symbols. In (G), individual transduced $Otof^{-/-}$ IHCs are displayed with thinner or broken red lines. In (B, D, E, G, H), data are displayed as mean ± s.e.m., ns $P$ > 0.05; **$P$ ≤ 0.01; ***$P$ ≤ 0.001 (Kruskal–Wallis test followed by Dunn's multiple comparison test).

otoferlin, which were absent in $Otof^{-/-}$ controls. However, due to a strong unspecific band at ~100 kDa, the presence of smaller otoferlin fragments that might interfere with the function of full-length otoferlin could not be excluded.

To quantify full-length otoferlin protein expression levels, we measured the fluorescence intensity of the C-terminal otoferlin antibody in transduced $Otof^{-/-}$ IHCs and normalized it to the C-terminal immunofluorescence in wild-type C57BL/6J (B6) IHCs (Figs 1E and EV3). Dual-AAV-mediated gene transfer into $Otof^{-/-}$ IHCs led to the expression of full-length otoferlin protein with 31–37% of wild-type levels (Fig 1E). We found the intracellular pattern of N- and C-terminal otoferlin immunolabeling in transduced $Otof^{-/-}$ IHCs to be similar to wild-type IHCs (Fig EV3). Non-transduced $Otof^{-/-}$ IHCs from contralateral non-injected ears displayed very weak N-terminal and C-terminal background otoferlin fluorescence signals of 3 ± 0.0% and 6 ± 0.0% of wild-type levels, respectively (Fig 1E, Appendix Fig S1 and S2). In dual-AAV-TS-transduced wild-type IHCs, the eGFP fluorescence signal was weaker than in those cells transduced with the single AAV2/6.eGFP virus (Figs EV1 and EV3).

An earlier study reported that $Otof^{-/-}$ IHCs have normal ribbon synapse numbers at P6, but only 60% of their synapses persist after P15 (Roux et al, 2006). By counting the number of CtBP2-labeled synaptic ribbons, we examined whether re-expressing otoferlin via dual-AAV vectors could reverse this loss (Fig 2A and B). At P26–29, $Otof^{-/-}$ IHCs from contralateral non-injected ears have 10 ± 0.3 synaptic ribbons ($n$ = 46 IHCs), similar to non-transduced $Otof^{-/-}$ IHCs in injected ears (9 ± 0.3 ribbons, $n$ = 65 IHCs). This is ~40% less than the 15 ± 0.3 synaptic ribbons in CD1B6F1 wild-type IHCs ($n$ = 108 IHCs) and 15 ± 0.5 synaptic ribbons in B6 wild-type IHCs ($n$ = 48 IHCs). Upon dual-AAV-mediated otoferlin expression, we found 10 ± 0.2 synaptic ribbons in dual-AAV-TS-transduced $Otof^{-/-}$ IHCs ($n$ = 59 IHCs) and 10 ± 0.3 synaptic ribbons in dual-AAV-Hyb-transduced $Otof^{-/-}$ IHCs ($n$ = 37 IHCs), which is virtually identical to the ribbon numbers in $Otof^{-/-}$ IHCs from contralateral non-injected ears. Thus, otoferlin gene delivery at P6–7 is apparently too late to prevent or reverse any synapse loss in $Otof^{-/-}$ IHCs. Accordingly, we re-assessed IHC synapse numbers in B6-$Otof^{-/-}$ and wild-type mice at P6 and P14 (Fig 2C and D). IHC synapses were identified as

CtBP2-labeled synaptic ribbons adjacent to Shank1a immunolabeled postsynaptic boutons (Huang et al, 2012). We found more synapses in P6 $Otof^{-/-}$ IHCs (49 ± 0.7 synapses; $n$ = 62 IHCs) than in wild-type IHCs (39 ± 0.3 synapses; $n$ = 53 IHCs). In the second postnatal week, synapse numbers decreased, reaching comparable numbers at P14 for $Otof^{-/-}$ mice (15 ± 0.3 synapses; $n$ = 65 IHCs) and wild-type mice (16 ± 0.2 synapses; $n$ = 73 IHCs, Fig 2D) pointing to a developmental delay in synapse loss compared to the $Otof^{-/-}$ mice assessed in Roux et al (2006). Notably, Shank1a labeled postsynaptic structures appeared larger in $Otof^{-/-}$ than in wild-type IHCs (Fig 2C). These findings, together with the failure to rescue synapse numbers by otoferlin re-expression from P6–7 on, point toward a yet undescribed role of otoferlin in IHC synapse maturation.

In rodents, SGNs diversify in the first four postnatal weeks into low, medium, and high spontaneous rate fibers to encode different sound intensities (Liberman & Liberman, 2016). In mature $Vglut3^{-/-}$ mice lacking ~40% of SGNs (Seal et al, 2008), almost all low spontaneous rate fibers and half of the medium rate fibers are missing (Shrestha et al, 2018). Similarly, aging or mild noise trauma result in the selective loss of low spontaneous rate fibers, which does not affect auditory thresholds, but likely impairs speech comprehension (Furman et al, 2013; Wu et al, 2018). It is currently unclear which type of auditory nerve fibers might be affected by the synapse loss in $Otof^{-/-}$ mice, whether synapse loss occurs at all in human patients and if so whether this would affect hearing after gene therapy. While SGN subtypes in $Otof^{-/-}$ mice could be identified by their molecular markers as in Shrestha et al (2018), future animal studies will be required to characterize spontaneous and sound-evoked firing rates of auditory nerve fibers in dual-AAV-treated $Otof^{-/-}$ mice. Co-administration of neurotrophic factors could additionally be tested to rescue these synapses (Wan et al, 2014; Suzuki et al, 2017).

By using cellular electrophysiological recordings, we investigated whether dual-AAV-mediated otoferlin gene transfer can rescue otoferlin-dependent fast exocytosis and synaptic vesicle replenishment in $Otof^{-/-}$ IHCs (Roux et al, 2006; Pangrsic et al, 2010; Fig 2E–H). Transduced IHCs in acutely isolated organs of Corti (P14–P18) were identified by their eGFP fluorescence and measured via perforated patch-clamp. Here, we found no difference in

voltage-gated $Ca^{2+}$-currents between dual-AAV-TS-transduced $Otof^{-/-}$ ($n = 8$ IHCs), non-transduced $Otof^{-/-}$ ($n = 10$ IHCs), and control CD1B6F1 wild-type IHCs ($n = 6$ IHCs, Fig 2E and H). $Ca^{2+}$-triggered exocytosis of the readily releasable synaptic vesicle pool (RRP) was measured as IHC plasma membrane capacitance increments ($\Delta C_m$) after short depolarization pulses (5–20 ms) eliciting $Ca^{2+}$ influx. Dual-AAV-mediated re-expression of otoferlin fully restored fast exocytosis of the RRP in $Otof^{-/-}$ IHCs (for 20 ms depolarization: wild-type CD1B6F1: $16.8 \pm 1.4$ fF, dual-AAV-TS-transduced $Otof^{-/-}$: $13.5 \pm 1.3$ fF; wild-type B6: $15.3 \pm 1.6$ fF (from Strenzke et al, 2016); $P = 0.3$, one-way ANOVA; non-transduced $Otof^{-/-}$: $1.4 \pm 0.8$ fF, $P = 1 \times 10^{-7}$, two-tailed t-test for transduced vs. non-transduced $Otof^{-/-}$ IHCs; Fig 2F and G). In agreement with our finding that around one out of four eGFP-fluorescent IHCs only expressed the N-terminal part of otoferlin (Figs 1D and EV3), we recorded two (out of ten) eGFP-expressing IHCs with hardly any $Ca^{2+}$-triggered exocytosis (broken red lines), similarly to non-transduced $Otof^{-/-}$ IHCs (Fig 2G). Presumably, the correct reassembly of the full-length otoferlin expression cassette in the right orientation did not take place in these two transduced IHCs. The lack of exocytosis in these cells is in agreement with earlier results (Reisinger et al, 2011), demonstrating that AAV-mediated co-expression of eGFP and synaptotagmin-1 neither rescued exocytosis in $Otof^{-/-}$ IHCs nor restored hearing in injected $Otof^{-/-}$ mice. Therefore, it can be ruled out that the AAV itself and/or the eGFP are able to recover exocytosis in the absence of otoferlin.

In contrast to the full rescue of fast exocytosis, sustained IHC exocytosis measured upon longer depolarization pulses (20–100 ms) and requiring otoferlin-dependent synaptic vesicle replenishment was only partially restored (Fig 2G). The estimated number of synaptic vesicles undergoing exocytosis between 20 and 100 ms of continuous depolarization in an average IHC synapse increased from $50 \pm 13$ synaptic vesicles/second/active zone (SV/s/AZ) in non-transduced $Otof^{-/-}$ ($n = 9$ IHCs) to $247 \pm 20$ SV/s/AZ in transduced $Otof^{-/-}$ IHCs ($n = 7$ IHCs) ($P < 0.0001$, two-tailed t-test). Still, these vesicle replenishment rates were 50 or 65% lower than in CD1B6F1 wild-type ($522 \pm 59$ SV/s/AZ, $n = 5$ IHCs) or B6 wild-type IHCs ($698 \pm 60$ SV/s/AZ, $n = 8$ IHCs) ($P = 0.0005$ for dual-AAV-TS vs. CD1B6F1 and $P < 0.0001$ for dual-AAV-TS vs. B6, Sidak's multiple comparisons test).

Otoferlin knock-out mice are profoundly deaf, and no auditory brainstem responses (ABRs) can be evoked by broadband click or tone burst sound stimuli (Roux et al, 2006; Pangrsic et al, 2010). Nevertheless, distortion product otoacoustic emissions (DPOAEs), measuring OHC amplification, and electrically evoked brainstem responses (eEBRs), measuring signal transmission through the afferent auditory pathway, are both normal in these mice and indicate that the impairment is specifically caused by a defect in the IHCs (Roux et al, 2006). We examined whether the dual-AAV strategy is able to restore auditory function in these animals by measuring ABRs in response to broadband clicks and tone bursts at different sound pressure levels (SPL) in 3- to 4-week-old (P23–30) dual-AAV-TS- ($n = 17$ mice) and dual-AAV-Hyb- ($n = 8$ mice) injected $Otof^{-/-}$ animals (Figs 3 and EV5). The characteristic ABR waveforms were partially rescued in all animals treated with otoferlin dual-AAV vectors (Figs 3A and B, and EV5A). Particularly, ABR waves II-V were clearly identifiable and served to determine ABR thresholds. In contrast, no ABRs could be elicited in non-treated

(-AAV) $Otof^{-/-}$ control littermates ($n = 38$ mice). Otoferlin dual-AAV injections restored auditory function with click-evoked ABR thresholds of $49 \pm 1$ dB SPL in dual-AAV-TS and $51 \pm 1$ dB SPL in dual-AAV-Hyb-transduced $Otof^{-/-}$ animals (range: 40–60 dB SPL; Fig 3C). In comparison, we measured click ABR thresholds of $32 \pm 2$ dB SPL in CD1B6F1 wild-type control animals injected with AAV2/6.eGFP ($n = 12$ mice) and $32 \pm 2$ dB SPL in otoferlin dual-AAV-TS-transduced wild-type mice ($n = 6$ mice; range: 20–40 dB SPL). In agreement with previous observations (Roux et al, 2006; Longo-Guess et al, 2007; Pangrsic et al, 2010; Reisinger et al, 2011), we recorded a prominent summating potential (SP) in non-treated $Otof^{-/-}$ mice evoked by the concerted depolarization of inner and outer HCs in response to broadband clicks (Figs 3A and B, and EV5A). The larger size of the SP compared to normal hearing animals is likely caused by the absence of efferent OHC inhibition, which requires auditory signal transmission along the afferent pathway (Fuchs & Lauer, 2018). In some dual-AAV-TS- and dual-AAV-Hyb-injected $Otof^{-/-}$ mice, the SP amplitude was similar to normal hearing mice, where AAV injection did not change the SP amplitude. Even though injection-induced HC loss cannot be excluded in $Otof^{-/-}$ mice, it seems plausible that the rescue of auditory signal transmission has activated the medial superior olivary nucleus (MSO), which hyperpolarizes OHCs by inhibitory synaptic contacts.

Tone burst ABR thresholds in dual-AAV-treated $Otof^{-/-}$ mice were generally much more variable between individual animals than click ABR thresholds (Figs 3D and EV5B). The discrepancy between consistently low ABR click thresholds compared to on average higher tone burst thresholds was also reported for $Vglut3^{-/-}$ mice rescued with a single AAV1 Vglut3-expressing virus (Akil et al, 2012). In our best mice, 8 and 12 kHz tone bursts of 50 dB SPL elicited ABRs (Figs 3D and EV5B). These frequencies are sensed in the apical half-turn of the mouse cochlea, where we found the highest IHC transduction rates (Fig 1D). The high variability can be explained as low SPL tone bursts induce a sharply tuned vibration of the basilar membrane only activating few IHCs that need to be transduced in order to evoke an ABR. Higher sound pressure levels cause a broader vibration peak of the basilar membrane and thus activate more IHCs, especially if the levels are high enough that OHC amplification is suppressed. In contrast, click sound stimuli contain a broadband of frequencies that activate IHCs along the entire cochlea, eliciting a measurable ABR as long as some IHCs have been transduced. In fact, we found no correlation between full-length otoferlin IHC transduction rates (entire cochlea, C-term otoferlin, Fig 1D) in dual-AAV-TS-treated animals and their individual click ABR thresholds ($n = 8$ mice; $r = -0.41$, $P = 0.5$, Spearman correlation test; Fig EV5C). Apparently, a click ABR threshold of ~50 dB can be established by very few dual-AAV-transduced IHCs. Improving dual-AAV vectors to gain higher otoferlin protein levels might result in even lower ABR thresholds, because synapses contacting low threshold SGNs might require a high vesicle replenishment rate, which correlates with the amount of otoferlin in IHCs (Strenzke et al, 2016).

Next, we quantified click ABR wave amplitudes in dual-AAV-treated $Otof^{-/-}$ mice, which were overall lower than in wild-type control mice (Figs 3E and EV5D–H). The ABR wave I amplitude, representing the summed activity of the auditory nerve, was very small and slightly delayed (Fig EV5D and I). This might be explained by the presence of fewer fully functional IHCs, fewer

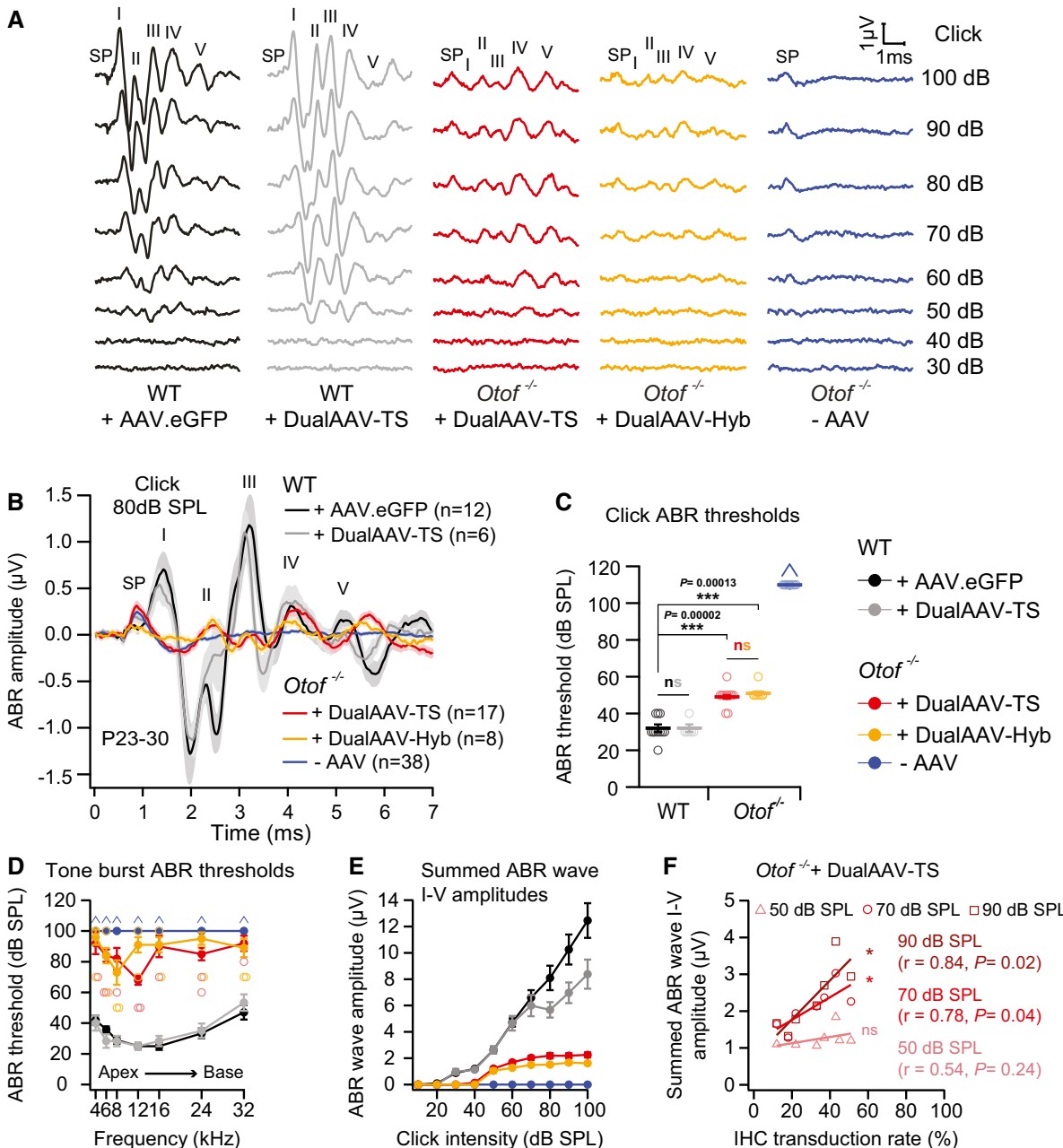

**Figure 3. Dual-AAV-mediated otoferlin gene delivery partially restores auditory function in deaf *Otof⁻/⁻* mice.**

A   Representative ABR wave traces in response to broadband click sound stimuli from otoferlin dual-AAV-TS (P26) and dual-AAV-Hyb (P27)-injected CD1B6F1-*Otof⁻/⁻* animals. AAV2/6.eGFP (+AAV.eGFP; P28) and dual-AAV-TS (P27)-injected CD1B6F1 wild-type, and non-injected control *Otof⁻/⁻* littermate (-AAV; P26) mice served as controls. SP: summating potential; ABR waves are indicated from I–V.

B   Average ABRs evoked by 80 dB SPL click sound stimuli for 20 clicks/s.

C, D   ABR click sound (C) and tone burst (D) thresholds in otoferlin dual-AAV-treated *Otof⁻/⁻* mice compared to wild-type and non-treated *Otof⁻/⁻* control animals. In (D), the two best animals are depicted with open circles. Animals with thresholds exceeding the maximum loudspeaker output (arrows) of 100 dB SPL for clicks and 90 dB SPL for tone bursts were set to 110 dB SPL and 100 dB SPL, respectively. Apical and basal cochlear turns are indicated as Apex and Base, respectively.

E   Summed ABR wave I–V amplitudes at different click sound intensities in otoferlin dual-AAV-injected, non-injected *Otof⁻/⁻*, and wild-type control mice.

F   Summed ABR wave I–V amplitudes of individual dual-AAV-TS-treated CD1B6F1-*Otof⁻/⁻* animals (*n* = 8 animals; from Fig 3E) plotted against their full-length otoferlin IHC transduction rates (from Fig 1D, C-term otoferlin). *r*: correlation coefficient.

Data information: In (B–F), age of analyzed animals: P23–30. In (B, C, E), number of analyzed mice: CD1B6F1 wild-type animals (+AAV.eGFP: *n* = 12 mice, dualAAV-TS: *n* = 6 mice) and CD1B6F1-*Otof⁻/⁻* animals (dualAAV-TS: *n* = 17 mice, dualAAV-Hyb: *n* = 8 mice, -AAV: *n* = 38 mice). In (D), number of analyzed mice is the same as for (B, C, E) except: CD1B6F1-*Otof⁻/⁻* animals (dualAAV-TS: *n* = 16 mice). In (B–E), data are represented as mean ± s.e.m. In (C, D, F), individual animals are depicted with open symbols. In (F), *r* ≥ 0.5 positive correlation (70 dB SPL and 90 dB SPL: Pearson correlation test; 50 dB SPL: Spearman correlation test). In (C), ns *P* > 0.05; *\*P* ≤ 0.05; \*\*\**P* ≤ 0.001 (Kruskal–Wallis test followed by Dunn's multiple comparison test).

synapses in these cells, and lower otoferlin protein levels in treated $Otof^{-/-}$ compared to wild-type mice, the latter most likely reducing spike synchrony (Buran *et al*, 2010; Bourien *et al*, 2014; Strenzke *et al*, 2016). The normal ABR wave II–IV latencies (Fig EV5I) and the increase in ABR wave amplitudes along the auditory pathway (Fig EV5D–H) suggest some degree of central auditory compensation of the peripheral sound encoding. Higher full-length otoferlin IHC transduction rates (entire cochlea, C-term otoferlin; Fig 1D) correlated with higher overall ABR wave amplitudes ($n = 8$ mice, $r = 0.78$ at 70 dB SPL and $r = 0.84$ at 90 dB SPL; $P = 0.040$ and $P = 0.017$, Pearson correlation test; Fig 3F). Consequently, we expect that increasing the full-length otoferlin IHCs transduction rate, e.g., by optimizing the injection procedure, will further improve ABR wave amplitudes.

Injecting AAV2/6.eGFP or otoferlin dual-AAV-TS vectors into the cochleae of wild-type mice had hardly any detrimental effects on auditory function when compared to non-treated control CD1B6F1 wild-type mice ($n = 6$ mice, average click threshold: $30 \pm 0$ dB SPL; Fig 3, Appendix Fig S4). No difference between injected and non-injected contralateral wild-type ears was observed in ABR wave I amplitude and wave I–V latencies (Appendix Fig S4C,E). Even though we did not observe an apparent HC loss in our immunohistochemical analyses and click ABR thresholds were unaffected, a minor elevation of the 24 kHz ABR threshold (Appendix Fig S4B) might point to potential damage of HCs by the injection. The sensory cells of the inner ear are particularly sensitive to pressure and volume changes that might occur while injecting the virus solution into the cochlea (Yoshimura *et al*, 2018). In addition, we found a reduction in ABR wave II–V amplitudes in injected compared to non-injected ears (Appendix Fig S4A,D), the origin of which is unclear.

It is interesting to note the similarities in otoferlin protein levels, sustained IHC exocytosis levels, ABR thresholds, and ABR wave amplitudes between dual-AAV-treated $Otof^{-/-}$ mice and the mildly hearing impaired $Otof^{I515T/I515T}$ mice (Strenzke *et al*, 2016). Hence, the treatment of DFNB9 patients with otoferlin dual-AAV vectors might result in almost normal auditory thresholds, but also in impaired speech comprehension and auditory fatigue similar to the hearing phenotype found in DFNB9 patients with the p.Ile515Thr mutation (Varga *et al*, 2006; Wynne *et al*, 2013). Since sustained exocytosis and auditory function seem to scale with otoferlin protein levels (Strenzke *et al*, 2016), we assume that increasing otoferlin protein expression levels via optimized dual-AAV half-vectors would consequently improve vesicle replenishment rates and cochlear function. In order to achieve this goal, the process of joining the two vector genomes in the correct orientation, the transcription efficiency, and the translation of the full-length transcript will need to be enhanced, e.g., by using codon-optimized cDNA. Nevertheless, improving the large transgene reconstitution efficacy by simply increasing the amount of available vector genomes does not improve protein expression levels as shown for dual-AAV vectors in the retina (Carvalho *et al*, 2017). Since the IHC transduction rates of our single AAV2/6 eGFP-expressing virus reached 99%, it is unlikely that other AAV serotypes such as AAV2/Anc80 (Landegger *et al*, 2017; Suzuki *et al*, 2017) will increase the amount of fully functional otoferlin protein levels in IHCs.

Even though dual-AAV-mediated otoferlin expression currently reaches only one third of wild-type protein levels, this might already be sufficient to substantially ameliorate the disease phenotype (e.g.,

improve speech comprehension) of *OTOF* patients with residual otoferlin expression. However, otoferlin protein fragments, caused by mutations inducing a premature STOP codon or resulting from incomplete reconstitution of the two split-AAV half-vectors, might compete with full-length otoferlin and consequently inhibit its function.

In conclusion, we provide a proof of concept that dual-AAV vector systems are suitable for the delivery of large transgenes such as otoferlin into mammalian IHCs and can at least partially restore auditory function. Further refinement of the split-AAV vectors, testing which AAV serotypes are most suitable for human application, and optimization of the injection procedure will pave the way for future gene replacement therapies of recessively inherited deafness forms like DFNB9.

# Materials and Methods

### Study approval

Animal handling and experiments complied with national animal care guidelines and were approved by the board for animal welfare of the University of Göttingen and the animal welfare office of the state of Lower Saxony, Germany.

### Animals

CD1xC57BL/6N-F1 (CD1B6F1) $Otof^{-/-}$ mice were obtained by crossing CD1-$Otof^{-/-}$ female with C57BL/6N-$Otof^{-/-}$ male mice (Reisinger *et al*, 2011). For wild-type controls, we cross-bred CD1 wild-type female mice (Charles River) with B6 wild-type ($Otof^{+/+}$) male mice, or used C57BL/6J mice (B6). Animals of both genders were used for all experiments. The mice were housed in social groups in individually ventilated cage (IVC) racks in a specific pathogen-free facility with free access to food and water and 12-h/ 12-h light/dark cycles.

### Dual-AAV constructs

Both otoferlin dual-5′AAV-TS and 5′AAV-Hyb half-vectors contained the human beta-actin promoter (hbA), a cytomegalovirus enhancer ($CMV_e$), eGFP, a P2A peptide sequence inducing ribosome skipping (Kim *et al*, 2011), the first half of the otoferlin coding sequence (CDS), and a splice donor sequence (SD; Trapani *et al*, 2014) at the 3′-end (Fig EV2). The dual-3′AAV-TS and 3′AAV-Hyb half-vectors contained a splice acceptor sequence (SA; Trapani *et al*, 2014) at the 5′-end, the second half of the otoferlin CDS, a woodchuck hepatitis virus post-transcriptional regulatory element (WPRE), and a bovine growth hormone polyadenylation sequence (pA). The dual-5′AAV-Hyb and 3′AAV-Hyb half-vectors additionally contained a highly recombinogenic sequence derived from the F1 phage (AK; Trapani *et al*, 2014) downstream the SD and upstream the SA site (Fig EV2). The mouse organ of Corti full-length otoferlin cDNA (transcript variant 4, KX060996; identical with NM_001313767; Strenzke *et al*, 2016) was split at the exon 21–exon 22 junction site into two halves each containing three otoferlin $C_2$-domains ($C_2A$-$C_2C$ in the dual-5′AAV half-vector and $C_2D$-$C_2F$ in the dual-3′AAV half-vector). To distinguish between the otoferlin dual-AAV cDNA and the wild-type cDNA, a silent point mutation, generating an

artificial restriction site (AccIII; TCCGGA), was introduced into the otoferlin CDS downstream the SA site (Fig EV4A).

### Virus production

The otoferlin dual-AAV constructs and an eGFP-expressing control construct, containing an hbA promoter, a $CMV_e$ enhancer, a WPRE element, and a pA sequence were packaged into an AAV2/6 sero-type. All viral vectors were produced by transient transfection of HEK293 cells in the presence of the helper plasmid pDP6. Viral particles were purified from cells harvested 2 days post-transfection by iodixanol step-gradient centrifugation and fast protein liquid chromatography (FPLC). Final virus preparations were dialyzed against PBS, and their purity was confirmed by SDS–gel electrophoresis. Virus genome (vg) titers were determined by qPCR.

### Postnatal injections

The virus solution was injected through the auditory bulla covering the round window membrane (RWM) into the scala tympani of the left cochlea at postnatal day 6–7 (P6–7) as described (Jung et al, 2015). All animals were anesthetized during the injection procedure via Isofluran (CONTRAfluran, ZeoSys) and locally anesthetized with Xylocain Pumpspray (AstraZeneca) before retroauricular incision. We used the following viral vectors for our experiments: AAV2/6.eGFP ($1.44 \times 10^{10}$ vg/µl), otoferlin dual-AAV2/6-TS half-vectors (1:1) ($1.2 \times 10^{10}$ vg/µl), and otoferlin dual-AAV2/6-Hyb half-vectors (1:1) ($1.38 \times 10^{10}$ vg/µl).

### RNA isolation, reverse transcription, PCR amplification, and sequencing

Total RNA was isolated from 3- to 4-week-old mice organs of Corti with the Invitrogen™ TRIzol™ Plus RNA Purification Kit (#12183555, Thermo Fisher Scientific) following the manufacturer's instructions. The SuperScript® IV First-Strand Synthesis System (#18091050, Thermo Fisher Scientific) served to synthesize cDNA. Otoferlin cDNA fragments spanning the split-site of the full-length otoferlin expression cassette were amplified from the cochlear cDNA using DreamTaq Polymerase (#EP0702, Thermo Fisher Scientific). The following primers were used for PCRs: 3′-CCCACAAGGC CAACGAGACGGATGAGGAC-5′ as forward and 3′-AAGAGGCTT CGGGCCTGATACATGTGTGCT-5′ as reverse primer. All PCR product bands were excised, TOPO TA cloned into a pCR2.1™-TOPO® vector using the TOPO® TA® Cloning Kit (#450641, Thermo Fisher Scientific), and transferred into One Shot™TOP10 Electro-comp™ E. coli cells (#C404050, Thermo Fisher Scientific). All clones were screened for the correct insert, and representative clones were subjected to Sanger sequencing.

### SDS–PAGE and Western blotting

Cochlear tissue from 3- to 4-week-old mice was lysed in RIPA buffer (50 mM Tris, 150 mM NaCl, 1% Triton X-100, 0.5% sodium deoxy-cholate, 0.1% SDS, pH 8.0) supplemented with protease inhibitors (cOmplete™, Mini, EDTA-free Protease Inhibitor Cocktail, Roche). Nuclei and cell debris were pelleted by low-speed centrifugation at 500 g for 5 min at 4°C, and the supernatant was used for Western

blotting. Protein concentration was determined with Pierce™ BCA Protein Assay Kit (#23227, Thermo Fisher Scientific).

Samples were separated by SDS–PAGE on NuPAGE™ 4–12% Bis–Tris Gels (Thermo Fisher Scientific) using Protein Marker VI (10–245) Prestained (Applichem USA) as a size marker and transferred onto nitrocellulose membranes (GE Healthcare Life Sciences). Membranes were probed with primary antibodies mouse IgG1 anti-otoferlin [13A9] (#ab53233, Abcam, 1:500) and rabbit anti-GAPDH (#247002, Synaptic Systems, 1:1,000) followed by incubation with secondary antibodies goat anti-mouse IgG-HRP (#115-035-146, Jackson ImmunoResearch, 1:2,000) and goat anti-rabbit IgG-HRP (#111-035-144, Jackson ImmunoResearch, 1:2,000). Pierce™ ECL Plus Western Blotting Substrate (#32132, Thermo Fisher Scientific) was used for detection.

### Patch-clamp electrophysiological recordings

Perforated patch-clamp recordings were used to measure $Ca^{2+}$-currents and plasma membrane capacitance increments ($\Delta C_m$) at room temperature in IHCs of acutely dissected apical turns of organs Corti (P14–18) as previously described (Moser & Beutner, 2000). The pipette solution contained the following: 130 mM Cs-gluconate, 10 mM tetraethylammonium chloride (TEA-Cl), 10 mM 4-amino-pyridine (Merck), 1 mM $MgCl_2$, 10 mM Cs-HEPES (pH 7.2, ~280 mOsm), and 300 µg/ml amphotericin B (Calbiochem). The extracellular solution contained the following: 110 mM NaCl, 35 mM TEA-Cl, 2.8 mM KCl, 2 mM $CaCl_2$, 1 mM $MgCl_2$, 10 mM Na-HEPES, 1 mM CsCl, and 11.1 mM D-glucose (pH 7.2, ~300 mOsm). All chemicals were purchased from Sigma-Aldrich, unless stated otherwise. An EPC-9 amplifier (HEKA Electronics) controlled by the Pulse software (HEKA Electronics) was used to filter low-pass currents at 5 kHz and sample them at 20 kHz. All recorded potentials were corrected for liquid-junction potentials (−14 mV). IHC recordings with a series resistance of > 30 MOhm or a leak conductivity of > 25 pS were excluded from analysis. Depolarization pulses were applied for different time periods with 30–60 s inter-stimulus intervals at peak $Ca^{2+}$-current potentials (usually at −14 mV) as described (Moser & Beutner, 2000). A P/6-protocol was used to correct for $Ca^{2+}$-current leak. The estimated average number of synaptic vesicles undergoing exocytosis during 0.08 ms (estimated average synaptic vesicle replenishment rate) was calculated from the $\Delta C_m$ difference between 100 ms and 20 ms depolarizations ($\Delta C_m 100$ ms − $\Delta C_m 20$ ms), 45 aF per vesicle (Neef et al, 2007), and the number of active zones per IHC at P14 according to the following equation:

$$\text{Vesicle replenishment rate } = [\Delta C_{m\,(100\,ms)} - \Delta C_{m\,(20\,ms)}]/0.08\,s/0.045\,fF/\text{synapse number.}$$

The number of active zones per IHC was determined by counting the CtBP2-immunolabeled synaptic ribbons adjacent to Shank1a-labeled postsynapses as depicted in Fig 2D (15 synapses/IHCs for $Otof^{-/-}$ IHCs and 16 synapses/IHCs for wild-type IHC).

### Immunohistochemistry

Cochlear whole mounts of injected and contralateral non-injected ears (P14–30) were immunostained as previously described

(Strenzke *et al*, 2016). The cochleae were perfused and fixed with 4% formaldehyde for 45 min at 4°C. Samples collected after ABR recordings from 3- to 4-week-old animals were additionally decalcified in Morse solution (10% sodium citrate, 22.5% formic acid) for 5 min or in 0.12 M EDTA (pH 8.0) for 2–3 days before the organs of Corti were dissected. The following primary antibodies were used: mouse IgG1 anti-otoferlin [13A9] (#ab53233, Abcam; 1:300) for labeling the N-terminal part of otoferlin, rabbit IgG anti-otoferlin (custom made; 1:100) for labeling the C-terminal part of otoferlin, goat IgG anti-calbindin D28K [C-20] (#sc-7691, Santa Cruz Biotechnology; 1:150) to visualize OHCs and IHCs, guinea pig anti-Vglut3 (#135204, Synaptic Systems; 1:300) to visualize IHCs, chicken IgY anti-GFP (#ab13970, Abcam, 1:500), rabbit IgG anti-Shank1a [C-terminal] (#RA19016, Neuromics; 1:300) for labeling the postsynaptic density, and goat IgG anti-CtBP2 [E-16] (#sc-5967, Santa Cruz Biotechnology; 1:100) and mouse IgG1 anti-CtBP2 [C-16] (#612044, BD Biosciences; 1:50) for labeling synaptic ribbons. The following secondary antibodies were used: Alexa Fluor 594- and Alexa Fluor 568-conjugated donkey anti-mouse IgG (#A21203, #A10037, Thermo Fisher Scientific, 1:200), Alexa Fluor 405-conjugated donkey anti-mouse IgG (#ab175658, Abcam, 1:200), Alexa Fluor 647- and Alexa Fluor 594-conjugated donkey anti-rabbit IgG (#A31573, #A21207, Thermo Fisher Scientific, 1:200), DyLight 405-conjugated donkey anti-goat IgG (#705-475-003, Jackson ImmunoResearch, 1:200), Alexa Fluor 594-conjugated donkey anti-goat IgG (#A11058, Thermo Fisher Scientific, 1:200), and Alexa Fluor 488-conjugated donkey anti-chicken IgY (#703-545-155, Jackson ImmunoResearch, 1:200). Confocal images were acquired using a laser scanning confocal microscope (Leica TCS SP5, Leica Microsystems GmbH) with a 10× air objective (NA = 0.40) for low and a 63× glycerol-immersion objective (NA = 1.3) for high magnifications images. Maximum intensity projections of optical confocal sections were generated using *ImageJ* (NIH, http://imagej.net/) and assembled in *Adobe Illustrator* (Adobe Systems).

The percentage of N-terminal and C-terminal otoferlin-positive IHCs were quantified from low magnification 3D images (3 μm z-stack step size) of apical, mid-basal, and basal turns of organs of Corti using the "Spots" tool in *Imaris 7.6.5* (Bitplane Scientific Software).

Protein expression levels were quantified from high magnification 3D IHC images (0.6 μm z-stack step size, 2× digital zoom) using a custom written *Matlab* (MathWorks) routine integrated into *Imaris 7.6.5* (Bitplane Scientific Software) as described (Strenzke *et al*, 2016).

Synaptic ribbon (immunolabeled with CtBP2) and synapse (co-immunolabeled with CtBP2 and Shank1a) numbers were counted from high magnification 3D IHC images (0.42 μm z-stack step size, 3× digital zoom) via the "Spots" tool in *Imaris 7.6.5* (Bitplane Scientific Software).

For IHC synapse and ribbon counting, images from apical cochlear turns were acquired, excluding the first 642.6 μm of the apex of the cochlea.

### Auditory brainstem response (ABR) recordings

Auditory brainstem responses were recorded from 3- to 4-week-old anesthetized mice subjected to 4, 6, 8, 12, 16, 24, 32 kHz tone burst (10 ms plateau, 1 ms $\cos^2$ rise/fall) or 0.03 ms broadband

### The paper explained

#### Problem

Congenital disabling hearing loss affects around one in 1000 newborns. Approximately, half of these cases are attributed to genetic causes in developed countries. More than 140 different deafness genes are known to date and more are expected to be characterized within the next years. Mutations in the gene *OTOF*, encoding the large protein otoferlin, lead to congenital recessive hearing loss DFNB9, addressed in this study. The acoustic signal transmission from auditory sensory hair cells to subsequent neurons is almost completely abolished in the absence of otoferlin. It is interesting to note that DFNB9 patients display no inner ear anomalies, which implies that reinstating the coding sequence of otoferlin into the cochlear sensory hair cells could restore hearing. However, a major challenge is the transfer of the long otoferlin coding sequence into these sensory cells. Viral vectors that can transport such large cDNAs are not suitable for gene delivery into hair cells and those vectors that are, have a limited transport capacity.

#### Results

In this study, the cargo limitation of adeno-associated viruses (AAVs) suitable for auditory sensory hair cell transduction was overcome by splitting otoferlin's cDNA into two halves and packaging them into two separate viruses. Co-injection of both viruses into cochleae of otoferlin knock-out mice led to the co-transduction of inner hair cells and to the reassembly and expression of the full-length mRNA and protein. Otoferlin protein levels reached about 30% of wild-type levels despite the predicted low efficiency of full-length transgene reassembly from two AAV half vectors. Fast inner hair cell exocytosis was completely restored and continuous vesicle replenishment partially recovered by the therapeutically reintroduced otoferlin. Auditory function in profoundly deaf otoferlin knock-out mice was partially rescued after split-AAV otoferlin treatment. Nonetheless, to ensure good speech comprehension in treated patients the split-AAV approach needs to be improved further to obtain higher otoferlin protein levels in auditory inner hair cells.

#### Impact

People with profound deafness usually receive cochlear implants, enabling them to understand spoken language, but have a limited hearing frequency resolution. Instead, gene therapy treatment has the potential to restore the full spectrum of hearing so that affected patients might be able to process vocal emotions, experience music, and understand speech in noisy environments like individuals without hearing impairment. This proof-of-concept study demonstrates that split viral vectors are suitable to partially restore hearing in otoferlin knock-out mice and, therefore, present a major step towards clinical gene therapy applications for this deafness form. Further optimization of these split-AAVs and their administration procedure might fully restore hearing in treated patients. Moreover, the split-AAV strategy can potentially rescue hearing in other forms of deafness caused by other large deafness genes.

click sound stimuli presented at 20 Hz as described (Jing *et al*, 2013). Injected ears were clogged with electrode gel while ABRs were recorded from contralateral non-injected ears. ABR click sound thresholds were determined as the lowest sound pressure levels necessary to evoke reproducible ABR wave responses and were measured in 10 dB SPL steps from 30 dB SPL to 100 dB SPL. Tone burst thresholds were recorded in 10 dB SPL steps from 10 dB SPL below the lowest reproducible ABR and up to 90 dB SPL. ABR wave I was defined as the first distinguishable peak between the summating receptor potential (SP) and the prominent ABR wave II peak. The amplitude of each ABR wave was

calculated as the difference between the highest point of a wave and the subsequent local minimum. The summed ABR wave I–V amplitude was calculated by adding up the individual amplitude values of ABR waves I–V.

## Statistics

We performed at least two independent experiments (e.g., animals) for our immunohistological analyses (e.g., synapse numbers in IHCs). For hearing function assessment, we measured at least 6 animals. For cellular electrophysiology, we recorded at least 6 cells, which originated from different animals and litters. All pups from a litter were treated the same way. We injected at least two litters for each condition/group, except for dual-AA-TS-treated wild-type mice, where only one litter was injected. All wild-type control and injected $Otof^{-/-}$ animals that showed reproducible ABR waveforms in response to click or tone burst sound stimuli were included into the analysis. However, around two thirds of injected $Otof^{-/-}$ animals showed neither any detectable ABR response nor transduced cells in immunostained organs of Corti at all, indicating that the virus solution did not enter into the cochlea. These animals were, thus, used as non-transduced $Otof^{-/-}$ controls.

Data averages are depicted as mean $\pm$ standard error of the mean (s.e.m.) and plotted using *Igor Pro 6* (WaveMetrics). Statistical analysis was performed via *GraphPad Prism 7.03* (GraphPad Software). The D'Agostino-Pearson omnibus and the Shapiro–Wilk tests were used to test for normality. The Brown–Forsythe test was used to test for equal variance in normally distributed data. The correlation coefficient ($r$) was calculated using the Pearson correlation test for parametric and the Spearman correlation test for non-parametric data (positive correlation $r \geq 0.5$; negative correlation $r \leq -0.5$). The Wilcoxon matched-pair signed rank test was used to test for statistical significance between two paired normally distributed data groups with unequal variance. The unpaired $t$-test with Welch's correction was used to test for statistical significance between two non-paired normally distributed data groups with unequal variance. The one-way ANOVA test followed by Tukey's or Sidak's multiple comparison test was used to test for statistical significance in parametric multiple comparisons. The Kruskal–Wallis test followed by the Dunn's multiple comparison test was used to test for statistical significance in non-parametric multiple comparisons (ns $P > 0.05$; *$P \leq 0.05$; **$P \leq 0.01$; ***$P \leq 0.001$). See Appendix Table S1 for exact $P$-values and statistical tests of individual experiments.

**Expanded View** for this article is available online.

## Acknowledgements

The authors would like to thank Stefan Thom, Nina-Katrin Dankenbrink-Werder, Sonja Heyrodt, and Monika Zebski for their excellent technical assistance. We are grateful to Christiane Senger-Freitag, Vladan Rankovic, and Nadine Dietrich for initial help with virus injections and ABR recordings. This work was supported by the University Medical Center Göttingen through a Heidenreich-von-Siebold fellowship to ER, the Göttingen Graduate School for Neurosciences, Biophysics, and Molecular Biosciences (GGNB) through a Junior Group Stipend to ER and HA-M, Akouos Inc., USA, the Deutsche Forschungsgemeinschaft (DFG) through the Collaborative Research Center 889, projects A2 (TM) and A4 (ER), and the Center for Nanoscale Microscopy and Physiology of the Brain (SK).

## Author contributions

ER, SK, and HA-M designed the study. HA-M, SK, ER, and APC performed experiments and analyzed data. ER, HA-M, and SK wrote the manuscript. All authors revised the manuscript. SYJ and TM established inner ear injections.

## Conflict of interest

The authors report being employees of University Medical Center Göttingen and co-inventors on a patent application for dual-AAV vectors to restore hearing. The University Medical Center Göttingen has licensed the rights to these parts of the patent exclusively to Akouos Inc., USA.

## For more information

(i)    https://omim.org/entry/601071.
(ii)   https://www.ncbi.nlm.nih.gov/books/NBK1251/.

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
