## [Review Process File · EMBO Molecular Medicine]

A dual-AAV approach restores fast exocytosis and partially rescues auditory function in deaf otoferlin knock-out mice

Hanan Al-Moyed, Andreia P. Cepeda, Sangyong Jung, Tobias Moser, Sebastian Kügler, Ellen Reisinger

Review timeline:

Submission date:	3 June 2018
Editorial Decision:	23 July 2018
Revision received:	4 October 2018
Editorial Decision:	26 October 2018
Revision received:	9 November 2018
Accepted:	12 November 2018

Editor: Céline Carret

Transaction Report:

1st Editorial Decision

23 July 2018

Thank you for the submission of your manuscript to EMBO Molecular Medicine. We have now heard back from the three referees whom we asked to evaluate your manuscript.

I am happy to report that all referees are enthusiastic about the study and supportive of publication. This said, a few controls are missing and should be added. Importantly, we would like you to address the AAV-eGFP in Otof-KO as commented by referee 2, along with providing additional details and clarifications.

We would welcome the submission of a revised version within three months for further consideration and would like to encourage you to address all the criticisms raised as suggested to improve conclusiveness and clarity. Please note that EMBO Molecular Medicine strongly supports a single round of revision and that, as acceptance or rejection of the manuscript will depend on another round of review, your responses should be as complete as possible.

I look forward to receiving your revised manuscript.

***** Reviewer's comments *****

Referee #1 (Remarks for Author):

In the present manuscript, Al-Moyed et al. assess a viral gene replacement strategy to rescue hearing in an inherited form of deafness. Because the target gene, otoferlin, extends beyond the capacity of conventional AAV, the authors explore the use of dual AAVs to deliver cDNA encoding for otoferlin to the otoferlin KO mouse (Otof^{-/-}). They assess two strategy: (1) trans-splicing (TS) uses artificial splice donor/acceptor before and after ITR in respective half vectors; and (2) dual hybrid (HYB) which includes a combination of overlapping vectors and splice donor/acceptors. Their work shows that both strategies lead to expression of full length transgene, recovery of fast but not sustained exocytosis in inner hair cells along with mild ABR recovery to broad band click stimuli. Interestingly, ABR response to pure tone stimuli appears variable and is mentioned but not illustrated in the manuscript.

While mild auditory sensitivity is observed after treatment with either vector, recovery of protein expression and fast exocytosis validates this novel dual AAV approach for the treatment of mutations associated with the OTOF gene. The manuscript is well written and the conclusions are appropriate. Some clarifications are required, however, that will strengthen this manuscript.

Major comments:

1- Since additional amplicons are observed in the Otof^{-/-} mice (Figure ev4b), question arises as to whether truncated proteins are expressed that may prevent full restoration of function by the transgene. Inclusion of a western blot confirming presence or absence of otoferlin protein transcripts would be very informative. At the very minimum, this point should be discussed.

2- Treatment with DualAAV-Hyb appears to be associated with reduction of summing potential (as seen in Figure 3). This point is not discussed but may be significant. Could there be hair cell loss associated with the DualAAV-Hyb treatment?

3- The use of pseudo colors in Figure 1 is interesting, however raw fluorescence images are needed to clarify the pattern of expression in particular with regard to GFP and Otoferlin (panel 1A).

4- Fig2a compares CTBP2 and Otoferlin immunostaining in DualAAV-Hyb and dualAAV-TS treated mice. It is unclear why C-term otoferlin antibody is used for TS while N-term otoferlin antibody is used for Hyb. Show same antibody for comparison, or both for each conditions.

5- Pg 7- First paragraph states that there was no evidence of IHC showing exclusively C-terminal otoferlin immunofluorescence, however, bar graph fig1 show more cells were labeled with C-term antibody in DualAAV-Hyb treated mice. Even though the difference does not appear to be significant, this result suggests that some cells are labeled with C-terminal otoferlin antibody only. This does seem surprising since there is no promoter driving expression of the C-termini. Please clarify and discuss.

6- Pg 12, figure Ev5b: IHC transduction rate is not correlated with click ABR threshold recovery. More of a comment here, but this result seems quite surprising. How were the transduction rates estimated? Was the count performed along the entire cochlea?

7- Since ABR response to pure tone stimuli were assessed and are described as "variable" in the text, an illustration would be useful that shows what frequencies were recovered. The manuscript does not differentiate apex versus base and questions arise as to where cellular recovery may be optimal.

Minor corrections:

1- Abstract, ln2: Define OTOF (first time it is introduced)

2- Pg 3: "Cochlear implant users often report experiencing speech understanding problems in noisy environments and can hardly perceive emotions in spoken language or music as a person without hearing impairment" - Please cite reference.

3- Pg 5, Results and discussion: Discuss why injections were performed at P6-P7 rather than P1-P2, a more commonly used time point for injections through the round window membrane.

4- Fig 2A, top left panel: which color is used for C-otoferlin?

5- EV1: Again different antibodies used in EV1A and EV1B; Show staining using the same antibody for consistency.

6- Pg 6: While restriction of otoferlin translation to auditory HCs may explain absence of otoferlin staining in transfected cells others than IHCs, other factors such as protein degradation could be involved. Include this in the discussion.

5. Fig 1S A, B: it is not clear what is depicted in the first column. If this is calbindin and GFP only, it should be clearly indicated. Also show raw fluorescent image only or along with pseudo-color.

Referee #2 (Comments on Novelty/Model System for Author):

DFNB9 is a human form of genetic deafness. The authors employ gene therapy in a mouse model of DFNB9 to rescue the auditory deficit. This work is a very nice proof-of-concept.

Referee #2 (Remarks for Author):

Review of the paper entitled "A dual-AAV approach restores fast exocytosis and partially rescues auditory function in deaf otoferlin knock-out mice" by Al-Moyed et al.

Mutations in OTOF, encoding otoferlin, cause non-syndromic recessive hearing loss. Experiments on *otof* knock-out and knock-in mutant mice revealed that this protein play an essential role in IHC exocytosis and vesicles replenishment, and is involved in vesicle reformation and endocytosis. One strategy to rescue this phenotype may be to transfer otoferlin cDNA into the IHC. Due to the large size of the transgene, such a transfer is difficult to accomplish. To solve this problem, the authors develop two different strategies: i.e. the trans-splicing (TC) and the "hybrid" strategy. Co-injection of the dual-AAV2/6, each containing one-half of the otoferlin cDNA into 6-7 day-old *otof*^{-/-} mice led to a full-length otoferlin mRNA. Although the loss of 40% of the synapses could not be prevented, dual-AAV transduction fully restore fast exocytosis, and replenishment reach 50 to 70% of the wild type values. Finally, click evoked brainstem responses at 40 -50 dB. Together, these results constitute the proof of concept that dual-AAV vectors can be used to deliver large transgenes such as otoferlin into the cochlear hair cells.

Major comments:

The paper is well written, and the results convincing, clearly described and merit to be published. Although the efficiency of the dual-AAV approach is impressive, a major hurdle is the lack of *Otof* KO transfected with AAV-eGFP as control for exocytosis and ABR. As it is, anyone can argue that the rescue is due to the AAV itself and not the *Otoferlin* re-introduction. I would therefore the authors to add these samples to strengthen their beautiful work.

Minor comments:

1. P. 6 "We presume that a yet unknown mechanism restricts the translation of otoferlin to auditory HCs." Still, do the authors may have some hypotheses?

2. P.8. "Earlier studies reported that *Otof*^{-/-} IHCs have normal ribbon synapse numbers at P6, but after the onset of hearing at P15 only 60% of synapses are left (Roux et al, 2006)." In this sentence "studies" call for several reports, but only the reference of Roux et al., 2006 is given. In addition, did the authors try to count in their experiments the number of ribbons at P6 to see whether Roux's data are confirmed (P6 IHC from *Otof* KO have normal synaptic ribbon number)?

3. P9. wild-type (n=6 IHCs) IHCs (Fig 2C, 2F). Please reshape, wild-type IHCs (n=6 IHCs ; Fig 2C, 2F)
4. P9. why mentioning high in the following sentence? "eliciting high Ca²⁺ influx"
5. P.10: second paragraph: The calculation of number of synaptic vesicles undergoing exocytosis per synapse between 20 and 100 ms of continuous depolarization is not a bit speculative? I understood that synapses in hair cells differ in their calcium channels number and hence in their synaptic release capacity. If true, the authors should be more cautious in their interpretation, and clearly mention that they proposed an "estimate" average number of vesicles.
6. P. 10: It would be good to write in the text the amount of secretion in ff the authors observed in control and AAV-treated mice.
7. Fig 2: legend and graph: Q_{real} should be replace by QCa²⁺
8. In the patch-clamp recordings (Fig.2), the authors compared WT secretion with transduced Otof KO. Should not the control include WT transduced with AAV-eGFP and more importantly Otof KO transduced with AAV-eGFP? From a logical point of view, we cannot conclude whether the rescue stems from the AAV alone or the AAV harboring the Otof construction?
9. In fig 2E, authors plot exocytosis from two WT strain (B16 and B16XCD1) but for Fig 2C, D and F the B16 data do not appear.
10. Fig.3.the reduction in the ABR amplitude following the DualAAV-TS transfection in WT need to be clearly mention and discuss.
11. Fig.3: The authors do not described the open circle in Fig.3C-D and we do not know what is summed ABR wave I-V amplitude: peak to peak, area under the curve? Please explain.
12. Fig.3. Since the authors do not report morphological data, I suggest at least to measure DPOAEs to insure that the treatment had no impact of the OHCs.
13. All the Figures: For a better understanding, authors should apply the same color code of the conditions to the legend text.

Referee #3 (Remarks for Author):

Nice paper extending inner ear gene therapy to inclusion of split AAVs. Good choice of analytical methods. Some issues with data and some need for language improvement, as detailed below.

Substantive comments

Fig 2 A, middle panels, show synapses in transduced (*) and non-transduced cells. My qualitative impression is that transduced cells have fewer synapses in mutant ears. Please add to the analysis in B, a count of synapses in transduced vs non-transduced cells in the same injected ear of mutants.

Comments about cochlear implant are inaccurate and unnecessary.

Appendix Figure S1 please include a higher resolution image taken with a x100 objective lens to better show the positive cells.

In images presented at higher magnification, both otof and GFP appear throughout cytoplasm and nucleus. Is this correct? Why?

Minor comments for language and clarity:

Yet, no large gene delivery method ambiguous (large gene or large method?)

prevents from administration of delete from

Dual-AAV mediated otoferlin expression is restricted to auditory hair cells. This statement needs improvement, as it stands, it can mean that OHCs are positive (they are not) and it can mean that vestibular hair cells are negative (data not shown).

Maximum confocal projections what does that mean?

Arrows need to point at the cell, not the vicinity.

no specific cell tropism.... Even within the organ of Corti I cannot see it in supporting cells, so the statement is unclear

with the images now provided I am unable to verify the statement: was also found in other cell types, e.g. SGNs

for explaining the change in ABRs after injection:...might result from damage caused by the injection procedure and/or the pressure increase in the cochlea during the injection what is the difference between the two?

Instead of can essentially restore auditory function How about can improve auditory function

1st Revision - authors' response

4 October 2018

***** Reviewer's comments *****

Referee #1 (Remarks for Author):

In the present manuscript, Al-Moyed et al. assess a viral gene replacement strategy to rescue hearing in an inherited form of deafness. Because the target gene, otoferlin, extends beyond the capacity of conventional AAV, the authors explore the use of dual AAVs to deliver cDNA encoding for otoferlin to the otoferlin KO mouse (*Otof*^{-/-}). They assess two strategy: (1) trans-splicing (TS) uses artificial splice donor/acceptor before and after ITR in respective half vectors; and (2) dual hybrid (HYB) which includes a combination of overlapping vectors and splice donor/acceptors. Their work shows that both strategies lead to expression of full length transgene, recovery of fast but not sustained exocytosis in inner hair cells along with mild ABR recovery to broad band click stimuli. Interestingly, ABR response to pure tone stimuli appears variable and is mentioned but not illustrated in the manuscript.

Thank you for your valuable comment. We added a graph as Figure EV 5B to illustrate the ABR variability in response to tone bursts.

While mild auditory sensitivity is observed after treatment with either vector, recovery of protein expression and fast exocytosis validates this novel dual AAV approach for the treatment of mutations associated with the OTOF gene. The manuscript is well written and the conclusion are appropriate. Some clarifications are required, however, that will strengthen this manuscript.

Major comments:

1- Since additional amplicons are observed in the *Otof*^{-/-} mice (Figure ev4b), question arises as to whether truncated proteins are express that may prevent full restoration of function by the transgene. Inclusion of a western blot confirming presence or absence of otoferlin protein transcripts would be very informative. At the very minimum, this point should be discussed.

We share the reviewer's view that protein fragments might interfere with the full-length otoferlin protein and inhibit its function. We have addressed this point in three ways:

- i) We included the sequencing results of these additional PCR amplicons obtained from *Otof*^{-/-} organs of Corti in Figure EV4C. The shortest fragment with 1379 bp contains the expected mRNA with a deletion of exons 14 and 15. The 1480 bp band contains, in addition, the intron between exons 20-21, and the 1679 bp fragment contains the intron between exons 23-24. The full alignment is available as separate file for the reviewer's interest.

- ii) The deletion of Exons 14 and 15 in our *Otof*^{-/-} mouse line (Reisinger *et al*, 2011) causes a frameshift and a premature stop, such that these additional splice variants could theoretically be translated into an otoferlin protein fragment. We performed a western blot for organ of Corti lysates from *Otof*-KO mice compared to wild-type mice, now added as Figure EV4D. Unfortunately, a strong band at around 100kDa found both in WT and KO samples precludes a clear analysis of protein fragments with a similar size. Nevertheless, we would like to show this western blot since it demonstrates the absence of full-length otoferlin (225kDa, pI/Mw tool, expasy.org) and another long otoferlin fragment (~210kDa) from our otoferlin knock-out. This is now described and discussed in the results/discussion section, page 8:
 In cDNA samples of dual-AAV injected and non-treated *Otof*^{-/-} organs of Corti, we amplified three otoferlin cDNA fragments of 1379bp, 1480 bp and 1679 bp, all lacking exons 14 and 15 (Fig EV4BC). The larger amplicons originate from incomplete splicing of the mutant mRNA (Fig EV4C). These splice variants might be translated into shorter fragments, the presence of which we assessed by western blot (Fig EV4D). In wild-type organs of Corti, we detected two specific bands of ~210-230 kDa, likely corresponding to full-length otoferlin, which were absent in *Otof*^{-/-} controls. However, due to a strong unspecific band at ~100 kDa, the presence of smaller otoferlin fragments that might interfere with the function of full-length otoferlin could not be excluded.
- iii) We discuss this point, also with respect to human mutations that might give rise to endogenously translated truncated otoferlin fragments, page 16:
 However, otoferlin protein fragments, caused by mutations inducing a premature STOP codon or resulting from incomplete reconstitution of the two split-AAV half vectors, might compete with full-length otoferlin and consequently inhibit its function.

2- Treatment with DualAAV-Hyb appears to be associated with reduction of summing potential (as seen in Figure 3). This point is not discussed but may be significant. Could there be hair cell loss associated with the DualAAV-Hyb treatment?

This is indeed an interesting observation, for which we provide more explanation in the revised version of the manuscript. In all otoferlin mouse mutants, the amplitude of the summing potential (SP) is much larger than in normal hearing control animals (Roux *et al*, 2006; Pangrsic *et al*, 2010; Reisinger *et al*, 2011). This is likely due to the absence of afferent activation, which in normal hearing animals induces efferent inhibition to the outer hair cells (Fuchs & Lauer, 2018). In our experiments, a partial rescue of IHC function leads to a reduction of the SP amplitude in some, but not all cases. We carefully checked individual animals treated with trans-splicing or hybrid dual-AAV vectors and found examples for large and small SP amplitudes for each approach (Response Fig 2 as an example for a small SP amplitude in a trans-splicing and a large amplitude in a hybrid injected mouse). In our immunohistochemical analysis, we did not see obvious signs of IHC or OHC loss (Appendix FigS2, FigS3), and thus we found no apparent correlation between reduced SP amplitudes and hair cell loss. Moreover, since wild-type mice injected with either eGFP or dual-AAV-TS did not display reduced SP amplitudes (Fig 3A, 3B, Appendix Fig S4A), we consider a major loss of hair cells due to the injection to be unlikely.

In conclusion, even though we cannot exclude a loss of hair cells caused by the injection procedure, we presume the reduction in SP amplitude is more likely attributed to the re-activated efferent inhibition. We included this point in the results/discussion, page 13:

The larger size of the SP compared to normal hearing animals is likely caused by the absence of efferent OHC inhibition, which requires auditory signal transmission along the afferent pathway (Fuchs & Lauer, 2018). In some dual-AAV-TS and dual-AAV-Hyb injected *Otof*^{-/-} mice, the SP amplitude was similar to normal hearing mice, where AAV-injection did not change the SP amplitude. Even though injection-induced HC loss cannot be excluded in *Otof*^{-/-} mice, it seems plausible that the rescue of auditory signal transmission has activated the medial superior olivary nucleus (MSO), which hyperpolarizes OHCs by inhibitory synaptic contacts.

Response Fig 2

Left panel: Individual ABR wave traces of a dual-AAV-TS injected mouse (red) with a strong reduction in the summing potential (SP) amplitude for stimuli between 30 and 90 dB.

Middle panel: Example ABR wave trace of a dual-AAV-Hyb injected *Otof*^{-/-} mouse (yellow) with a prominent SP amplitude comparable to non-treated *Otof*^{-/-} ABR waves (right panel, blue).

Examples for a dual-AAV-TS injected mouse with a prominent SP amplitude and a dual-AAV-Hyb injected animal with a small SP amplitude are found in the main manuscript, Fig 3A.

3- The use of pseudo colors in Figure 1 is interesting, however raw fluorescence images are needed to clarify the pattern of expression in particular with regard to GFP and Otoferlin (panel 1A).

Done, see Appendix Fig S3A.

We prefer the use of “fire” lookup table color for the figures in the main manuscript since we are afraid that weak immunofluorescence signals might be hard to recognize in the raw fluorescence images.

4- Fig2a compares CTBP2 and Otoferlin immunostaining in DualAAV-Hyb and dualAAV-TS treated mice. It is unclear why C-term otoferlin antibody is used for TS while N-term otoferlin antibody is used for Hyb. Show same antibody for comparison, or both for each conditions.

We apologize for displaying immunofluorescence from different antibodies here, which is due to technical reasons only and is supposed to have no biological relevance in this case. In this experiment, otoferlin immunolabelling was solely used to identify dual-AAV transduced cells. For the N-terminal antibody, we omitted cells with weak otoferlin fluorescence from the analysis, since those could be the ones with no split-AAV re-assembly. The otoferlin N-terminal antibody is a mouse antibody and the C-terminal is a rabbit polyclonal one. For synaptic ribbon immunostainings as shown in Fig 2A-B we performed a post-synaptic co-labelling in parallel, which did not work for all of these immunostainings, unfortunately. Depending on the antibodies used for such co-labellings, we took one or the other otoferlin antibody.

5- Pg 7- First paragraph states that there was no evidence of IHC showing exclusively C-terminal otoferlin immunofluorescence, however, bar graph fig1 show more cells were labeled with C-term antibody in DualAAV-Hyb treated mice. Eventhough the difference does not appear to be significant, this result suggests that some cells are labeled with C-terminal otoferlin antibody only. This does seem surprising since there is no promoter driving expression of the C-termini. Please clarify and discuss.

We are sorry for not being clear and causing confusion. The bar graph in Fig. 1E does NOT display the number of IHCs labelled with one or the other otoferlin antibody, but it shows the relative otoferlin immunofluorescence intensity levels of each antibody for only otoferlin positive IHCs normalized to B6 wild-type IHC immunofluorescence intensity levels of the same antibody. In contrast, the graph in Fig 1D displays the number of otoferlin-positive IHCs, where the number of C-terminally labelled IHCs is always lower than the number of N-terminally labelled IHCs.

We revised the text and added the quantification of immunofluorescence background levels in IHCs from non-transduced *Otof*^{-/-} contralateral organs of Corti to Fig 1E.

To quantify full-length otoferlin protein expression levels, we measured the fluorescence intensity of the C-terminal otoferlin antibody in transduced *Otof*^{-/-} IHCs and normalized it to the C-terminal immunofluorescence in wild-type C57BL/6J (B6) IHCs (Fig 1E, Fig EV3).

6- Pg 12, figure Ev5b: IHC transduction rate is not correlated with click ABR threshold recovery. More of a comment here, but this result seems quite surprising. How were the transduction rates estimated? Was the count performed along the entire cochlea?

The number of otoferlin C-terminal labelled IHCs was counted along the entire cochlea in injected ears and divided by the number of calbindin immunolabelled IHCs. The number of the C-terminal otoferlin positive IHCs for individual animals (n= 8 mice) used for the correlation analysis in Fig EV5C are shown in Fig 1D and their individual ABR thresholds are depicted in Fig 3C.

We included the following paragraph in the results and discussion part of the manuscript to better clarify this point (page 14):

In fact, we found no correlation between full-length otoferlin IHC transduction rates (entire cochlea, C-term otoferlin, Fig 1D) in dual-AAV-TS treated animals and their individual click ABR thresholds (n= 8 mice; $r = -0.41$, $P = 0.5$, Spearman correlation test; Fig EV5C). Apparently, a click ABR threshold of ~50dB can be established by very few dual-AAV transduced IHCs. Improving dual-AAV vectors to gain higher otoferlin protein levels might result in even lower ABR thresholds, because synapses contacting low threshold SGNs require a high vesicle replenishment rate, which correlates with the amount of otoferlin in IHCs (Strenzke *et al*, 2016).

7- Since ABR response to pure tone stimuli were assessed and are described as "variable" in the text, an illustration would be useful that shows what frequencies were recovered. The manuscript does not differentiate apex versus base and questions arise as to where cellular recovery may be optimal.

We have now included a separate graph (Fig EV5B) that illustrates tone burst ABR thresholds measured from individual animals to better display the variability between these mice. The best threshold recovery was found at 8-12 kHz, which is in agreement with our finding that full-length otoferlin IHC transduction rates were generally higher in the apical cochlear turn as seen in Fig 1D. Note that we also counted otoferlin positive dual-AAV transduced IHCs for the apex and the base of the cochlea (Fig. 1D).

We included the following paragraph in the results and discussion part of the manuscript to better clarify this point (page 13):

In our best mice, 8 and 12kHz tone bursts of 50 dB SPL elicited ABRs (Fig 3D, Fig EV5B). These frequencies are sensed in the apical half-turn of the mouse cochlea, where we found the highest IHC transduction rates (Fig 1D).

Minor corrections:

1- Abstract, ln2: Define OTOF (first time it is introduced)
Done.

2- Pg 3: "Cochlear implant users often report experiencing speech understanding problems in noisy environments and can hardly perceive emotions in spoken language or music as a person without hearing impairment" - Please cite reference.

We slightly revised the sentence and included the following references:

Cochlear implant users report difficulties in speech understanding during noise and in perceiving vocal emotions, and typically cannot experience music as a person without hearing impairment (Fu *et al*, 1998; Nelson *et al*, 2003; Oxenham & Kreft, 2014; Luo *et al*, 2007; Most & Aviner, 2009; Chatterjee *et al*, 2015; Paquette *et al*, 2018; McDermott, 2004).

3- Pg 5, Results and discussion: Discuss why injections were performed at P6-P7 rather than P1-P2, a more commonly used time point for injections through the round window membrane.

With respect to a future gene therapy in humans, we aimed for a rather late injection time point.

We added this in the main manuscript at page 5.:

We aimed for a rather late time point of treatment, since the early development of the inner ear does not seem to require otoferlin (Roux *et al*, 2006), making gene therapy of mature *Otof*^{-/-} IHCs feasible in theory. AAVs were injected into the cochlea at postnatal day 6-7 (P6-7) because the auditory bulla structure covering the round window membrane (RWM) is still soft enough at this developmental stage to be penetrated well with an injection glass pipette.

4- Fig 2A, top left panel: which color is used for C-otoferlin?

Magenta. The color “magenta” is used instead of the color “red” to consider color-blind readers as stated in the general EMBO “author guidelines”.

5- EV1: Again different antibodies used in EV1A and EV1B; Show staining using the same antibody for consistency.

Here, the otoferlin antibody serves only for the identification of inner hair cells in wild-type organs of Corti. The same could be done with an antibody against Vglut3 for instance. The experiment has been done on wild-type mice, where all IHCs are otoferlin positive. Figure EV1 aims to illustrate that both of our control eGFP expressing AAV2/6 (panel A) and the dual-AAV2/6 (panel B) viruses transduce several cell types (spiral ganglion neurons, inner hair cells, supporting cells), as indicated by eGFP fluorescence in hair cells (B1) and in spiral ganglion neurons (B2).

We revised the figure legend for clarification.

Figure EV 1 - AAV2/6 transduces various cell types in the inner ear.

A,B Low magnification views for eGFP immunofluorescence in CD1B6F1 wild-type organs of Corti transduced with AAV2/6 vectors, indicating a broad cell type tropism both for a single eGFP expressing AAV2/6 (**A**; P23) and eGFP expressed from otoferlin dual-AAV-TS vectors (**B**; P27).

6- Pg 6: While restriction of otoferlin translation to auditory HCs may explain absence of otoferlin staining in transfected cells others than IHCs, other factors such as protein degradation could be involved. Include this in the discussion.

Done, page 6:

We presume that a yet unknown mechanism restricts the expression of otoferlin to auditory HCs, such as post-transcriptional regulation or protein degradation.

7. Fig 1S A, B: it is not clear what is depicted in the first column. If this is calbindin and GFP only, it should be clearly indicated. Also show raw fluorescent image only or along with pseudo-color.

We revised the figure legend for clarification.

The first column shows all 4 immunostainings: Calbindin (blue), eGFP (green), N-terminal otoferlin (magenta), and C-terminal otoferlin (white).

We have prepared an additional Figure showing the same organs of Corti displayed in Appendix Fig S1A-B, but with the raw fluorescent images instead of the “fire” color lookup tables (Response Fig. 3).

**Response Fig 3**

Raw fluorescent images corresponding to Appendix Fig S1.

Referee #2 (Comments on Novelty/Model System for Author):

DFNB9 is a human form of genetic deafness. The authors employ gene therapy in a mouse model of DFNB9 to rescue the auditory deficit. This work is a very nice proof-of-concept.

Referee #2 (Remarks for Author):

Review of the paper entitled "A dual-AAV approach restores fast exocytosis and partially rescues auditory function in deaf otoferlin knock-out mice" by Al-Moyed et al.

Mutations in OTOF, encoding otoferlin, cause non-syndromic recessive hearing loss. Experiments on *otof* knock-out and knock-in mutant mice revealed that this protein play an essential role in IHC exocytosis and vesicles replenishment, and is involved in vesicle reformation and endocytosis. One strategy to rescue this phenotype may be to transfer otoferlin cDNA into the IHC. Due to the large size of the transgene, such a transfer is difficult to accomplish. To solve this problem, the authors develop two different strategies: i.e. the trans-splicing (TS) and the "hybrid" strategy. Co-injection of the dual-AAV2/6, each containing one-half of the otoferlin cDNA into 6-7 day-old *otof*^{-/-} mice

led to a full-length otoferlin mRNA. Although the loss of 40% of the synapses could not be prevented, dual-AAV transduction fully restore fast exocytosis, and replenishment reach 50 to 70% of the wild type values. Finally, click evoked brainstem responses at 40 -50 dB. Together, these results constitute the proof of concept that dual-AAV vectors can be used to deliver large transgenes such as otoferlin into the cochlear hair cells.

Major comments:

The paper is well written, and the results convincing, clearly described and merit to be published. Although the efficiency of the dual-AAV approach is impressive, a major hurdle is the lack of Otof KO transfected with AAV-eGFP as control for exocytosis and ABR. As it is, anyone can argue that the rescue is due to the AAV itself and not the Otoferlin re-introduction. I would therefore the authors to add these samples to strengthen their beautiful work.

We understand that the reviewer is afraid that the AAV itself or the eGFP might be able to rescue IHC exocytosis and hearing in *Otof*^{-/-} mice. In a previous study from our lab (Reisinger et al., 2011), we showed that the AAV-mediated expression of eGFP (coexpressed with synaptotagmin-1) does neither rescue exocytosis in otoferlin-knock-out inner hair cells (Reisinger et al, 2011, Figure 3) nor hearing in otoferlin knock-out animals (Reisinger et al., 2011, Figure 2). In the same study, we also found that AAV-mediated expression of synaptotagmin-1 and eGFP do not inhibit exocytosis in wild-type inner hair cells or affect hearing. We feel that these previous results unambiguously demonstrate that transduction of IHCs by an AAV vector itself has no influence on exocytosis and hearing in otoferlin knock-out mice.

In the revised version of the manuscript, we added this information and related it to the finding of the two eGFP fluorescent cells with hardly any exocytosis (page 11):

In agreement with our finding that around one out of four eGFP-fluorescent IHCs only expressed the N-terminal part of otoferlin (Fig 1D and EV3), we recorded two (out of 10) eGFP-expressing IHCs with hardly any Ca²⁺-triggered exocytosis (dashed lines), similarly to non-transduced *Otof*^{-/-} IHCs (Fig 2G). Presumably, the correct reassembly of the full-length otoferlin expression cassette in the right orientation did not take place in these two transduced IHCs. The lack of exocytosis in these cells is in agreement with earlier results (Reisinger *et al*, 2011), demonstrating that AAV mediated co-expression of eGFP and synaptotagmin-1 did neither rescue exocytosis in *Otof*^{-/-} IHCs nor restore hearing in injected *Otof*^{-/-} mice. Therefore, it can be ruled out that the AAV itself and/or the eGFP are able to recover exocytosis in the absence of otoferlin.

Minor comments:

1. P. 6 "We presume that a yet unknown mechanism restricts the translation of otoferlin to auditory HCs." Still, do the authors may have some hypotheses?

This is an interesting finding for which we currently do not have a straightforward explanation.

Future work will be required to narrow down the mechanisms that might be involved in this process. In accordance with the comment of referee 1 (point 6), we now mention two potential hypotheses in the revised manuscript (page 7):

We presume that a yet unknown mechanism, such as post-transcriptional regulation or targeted protein degradation, restricts the expression of otoferlin to auditory HCs.

2. P.8. "Earlier studies reported that *Otof*^{-/-} IHCs have normal ribbon synapse numbers at P6, but after the onset of hearing at P15 only 60% of synapses are left (Roux et al, 2006)." In this sentence "studies" call for several reports, but only the reference of Roux et al., 2006 is given. In addition, did the authors try to count in their experiments the number of ribbons at P6 to see whether Roux's data are confirmed (P6 IHC from *Otof* KO have normal synaptic ribbon number)?

Thank you for this valuable comment. We corrected the respective sentence to "An earlier study...".

We re-assessed the synapse numbers at P6 and P14 in *Otof*^{-/-} and compared them to wild-type control organs of Corti. In contrast to Roux et al.(2006), we found more synapses in *Otof*^{-/-} IHCs at P6 than in wild-type IHCs. At P14, the number of synapses was similar in *Otof*^{-/-} IHCs and in wild-type IHCs. Our results indicate that *Otof*^{-/-} IHCs seem to lose more synapses in the second and third postnatal week than wildtype IHCs. Importantly, these findings indicate that early postnatal synapse development appears to be different in the absence of otoferlin, which is in line with our finding that dual-AAV mediated expression of otoferlin from P6-7 onwards is apparently too late to prevent the loss of 40% of synapses. Thus, we now have two lines of evidence that otoferlin, contrary to the previous report, seems to be required for synaptic maturation. Furthermore, the novel P14 synapse

counts provide an explanation why Ca^{2+} currents did not differ between *Otof*^{-/-} and wild-type IHCs when measured between P14-18. We used the new synapse counts at P14 to calculate the vesicle replenishment rates.

We included these novel findings in the abstract, in Fig 2C-D and the main text of the manuscript (page 9-10).

Accordingly, we re-assessed IHC synapse numbers in CD1B6F1 *Otof*^{-/-} and wild-type mice at P6 and P14 (Fig 2C,D). IHC synapses were identified as CtBP2-labelled synaptic ribbons adjacent to Shank1a immunolabeled postsynaptic boutons (Huang *et al*, 2012). We found more synapses in P6 *Otof*^{-/-} IHCs (49±0.7 synapses; n=62 IHCs) than in wild-type IHCs (39±0.3 synapses; n=53 IHCs). In the second postnatal week, synapse numbers decreased, reaching comparable numbers at P14 for *Otof*^{-/-} (15±0.3 synapses; n=65 IHCs) and wild-type mice (16±0.2 synapses; n=73 IHCs, Fig 2D). Notably, Shank1a labelled postsynaptic structures appeared larger in *Otof*^{-/-} than in wild-type IHCs (Fig 2C). These findings, together with the failure to rescue synapse numbers by otoferlin re-expression from P6-7 on, point towards a yet undescribed role of otoferlin in IHC synapse maturation.

3. P9. wild-type (n=6 IHCs) IHCs (Fig 2C, 2F). Please reshape, wild-type IHCs (n=6 IHCs ; Fig 2C, 2F)

Done

4. P9. why mentioning high in the following sentence? "eliciting high Ca^{2+} influx
"high" was deleted from this sentence

5. P.10: second paragraph: The calculation of number of synaptic vesicles undergoing exocytosis per synapse between 20 and 100 ms of continuous depolarization is not a bit speculative? I understood that synapses in hair cells differ in their calcium channels number and hence in their synaptic release capacity. If true, the authors should be more cautious in their interpretation, and clearly mention that they proposed an "estimate" average number of vesicles.

We agree and revised the sentence accordingly:

The estimated number of synaptic vesicles undergoing exocytosis between 20 and 100 ms of continuous depolarization on an average synapse increased from...

6. P. 10: It would be good to write in the text the amount of secretion in FF the authors observed in control and AAV-treated mice.

Done

7. Fig 2: legend and graph: Qreal should be replace by QCa^{2+}

Done

8. In the patch-clamp recordings (Fig.2), the authors compared WT secretion with transduced *Otof* KO. Should not the control include WT transduced with AAV-eGFP and more importantly *Otof* KO transduced with AAV-eGFP? From a logical point of view, we cannot conclude whether the rescue stems from the AAV alone or the AAV harboring the *Otof* construction?

As indicated above, the requested control experiments have been performed in an earlier study (Reisinger *et al.*, 2011).

The AAV consists of capsid proteins required only during cell entry and are degraded further on. The capsid proteins surround the AAV genome, which is a single stranded DNA molecule. The AAV genome is flanked by two non-coding 145 nucleotide-long inverted terminal repeats (ITRs) that form tail-to-head concatemers of several AAV genomes in target cells (Yan *et al*, 2005). All other elements of the original AAV genome are replaced in these recombinant AAVs by defined sequences such as promoter, coding sequence of the gene of interest, an mRNA stabilizing element (e.g. WRPE), and a polyadenylation signal. In Reisinger *et al.*, 2011 we showed that an AAV encoding eGFP and synaptotagmin-1 did neither rescue exocytosis in *Otof*^{-/-} inner hair cells nor did it restore hearing. We can, therefore, rule out that the AAV itself or the eGFP are able to restore exocytosis in the absence of otoferlin. In addition, in Reisinger *et al.*, 2011 we proved that expression of eGFP and synaptotagmin-1 in wild-type inner hair cells did not have any effect on exocytosis or hearing.

9. In fig 2E, authors plot exocytosis from two WT strain (B16 and B16XCD1) but for Fig 2C, D and F the B16 data do not appear.

The B6 data were replotted from Strenzke et al., 2016, which we now indicated more clearly to show a comparison of our rescue data with an additional representative wild-type dataset. Since the Ca^{2+} peak currents (now Fig. 2E) and the Ca^{2+} current integrals (now Fig. 2H) are not different in the three presented groups, we felt that a comparison to more wild-type B6 data is not necessary. Fig2D (now Fig 2F), only shows representative example traces. For this reason we have only displayed the B6 data in panel E (now Fig 2G), where we found significant differences between the three groups.

10. Fig.3.the reduction in the ABR amplitude following the DualAAV-TS transfection in WT need to be clearly mention and discuss.

We now explicitly mention and discuss the impact of postnatal round window membrane AAV-injections on hearing in WT animals. Please note that ABR wave I amplitude was unchanged between dual-AAV-TS injected wild-type ears and contralateral non-injected ears (Appendix Fig S4C).

Page 15:

No difference between injected and non-injected contralateral wild-type ears was observed in ABR wave I amplitude and wave I-V latencies (Appendix Fig S4C,E). Even though we did not observe an apparent HC loss in our immunohistochemical analyses and click ABR thresholds were unaffected, a minor elevation of the 24kHz ABR threshold (Appendix Fig S4B) might point to a potential damage of HCs by the injection. The sensory cells of the inner ear are particularly sensitive to pressure and volume changes that might occur while injecting the virus solution into the cochlea (Yoshimura *et al*, 2018). In addition, we found a reduction in ABR wave II-V amplitudes in injected compared to non-injected ears (Appendix Fig S4A, D), the origin of which is unclear.

11. Fig.3: The authors do not described the open circle in Fig.3C-D and we do not know what is summed ABR wave I-V amplitude: peak to peak, area under the curve? Please explain.

The open circles are explained in the “data information” section at the end of the figure legend, and represent data from individual animals. The amplitudes of each wave were determined as the distance from the local maximum to the next local minimum (peak-to-valley). The summed ABR wave I-V amplitude is the sum of the individual peak-to-valley amplitudes of ABR waves I-V. We have added this description to our “Materials & Methods” section on page 8.

12. Fig.3. Since the authors do not report morphological data, I suggest at least to measure DPOAEs to insure that the treatment had no impact of the OHCs.

We did not observe any obvious OHC loss in the immunostainings of transduced organs of Corti compared to non-injected contralateral ears as seen in Appendix Fig S1, S2, S3. In all our controls, the presence of OHC and IHCs was confirmed by immunostaining against calbindin (blue). We did not perform DPOAE recordings in the first place since those are wild-type like in *Otof*^{-/-} mice, thus we did not expect a direct effect of the dual-AAV treatment on outer hair cell function. Moreover, a minor loss of OHCs, as it can be caused by RWM injections, does not necessarily affect DPOAEs, as e.g. in Landegger et al., (2017), Fig. 2g. Our revised results and discussion section now explicitly mentions the possibility of OHC damage, see response to point 10.

13. All the Figures: For a better understanding, authors should apply the same color code of the conditions to the legend text.

We have now indicated the color “white” in the figure legend for immunolabels in white. For other colors, we have not included a color code description in the manuscript and figure legends to comply with EMBO’s author guidelines. This point can be changed if requested by the editors.

Referee #3 (Remarks for Author):

Nice paper extending inner ear gene therapy to inclusion of split AAVs. Good choice of analytical methods. Some issues with data and some need for language improvement, as detailed below.

Substantive comments

Fig 2 A, middle panels, show synapses in transduced (*) and non-transduced cells. My qualitative impression is that transduced cells have fewer synapses in mutant ears. Please add to the analysis in B, a count of synapses in transduced vs non-transduced cells in the same injected ear of mutants.

Done. See Fig 2B in the revised manuscript. The number of synaptic ribbons was the same in all IHC with *Otof*^{-/-} background.

Comments about cochlear implant are inaccurate and unnecessary.

With respect to the comment of reviewer 1, we revised the sentences and added references that specifically support the current statement. We think it is of importance to discuss the potential benefits of gene therapy over cochlear implantation at this point.

Given the normal inner ear morphology in these patients, a postnatal transfer of otoferlin cDNA into the inner ear is predicted to ameliorate this hearing loss. Thus, gene therapy might outperform the otherwise necessary cochlear implantation, which transmits only part of the acoustic information. Cochlear implant users report difficulties in speech understanding during noise and in perceiving vocal emotions, and typically cannot experience music as a person without hearing impairment (Fu *et al*, 1998; Nelson *et al*, 2003; Oxenham & Kreft, 2014; Luo *et al*, 2007; Most & Aviner, 2009; Chatterjee *et al*, 2015; Paquette *et al*, 2018; McDermott, 2004).

Appendix Figure S1 please include a higher resolution image taken with a x100 objective lens to better show the positive cells.

We prepared a new figure (Appendix Fig. S2) showing higher magnification views of the transduced organs of Corti displayed in Appendix Fig S1. Please see also Figure EV3 for higher resolution images of transduced cells.

In images presented at higher magnification, both *otof* and GFP appear throughout cytoplasm and nucleus. Is this correct? Why?

GFP is indeed found in the cytoplasm and the nucleus of IHCs. It is a soluble protein which can enter the nucleus of any cell (Seibel *et al*, 2007). Otoferlin is typically localized to the plasma membrane and is distributed throughout the cytoplasm of inner hair cells except the most apical subcuticular space. It is not present in the nucleus.

For better illustration, we prepared a figure (Response Fig 4) showing maximum intensity projections with only 4-5 optical confocal sections of the IHCs displayed in Fig EV3.:

Response Fig 4

Maximum intensity projections of 4-5 optical confocal sections showing the distribution of eGFP (upper panel), otoferlin (middle panels), and calbindin. EGFP is present in the nuclei and cytoplasm of IHCs, while otoferlin is absent from the nuclei of the cells. Otoferlin and calbindin are present in the cytoplasm. In addition, otoferlin localizes to the basolateral plasma membrane. Individual stainings are depicted as color lookup tables. Scale bars: 10 μ m

Minor comments for language and clarity:

Yet, no large gene delivery method ambiguous (large gene or large method?)

Rephrased to: “no delivery method for large genes”

prevents from administration of; delete “from”

Done

Dual-AAV mediated otoferlin expression is restricted to auditory hair cells. This statement needs improvement, as it stands, it can mean that OHCs are positive (they are not) and it can mean that vestibular hair cells are negative (data not shown).

Occasionally, we see dual-AAV mediated otoferlin expression in outer hair cells (see Appendix Fig. S3), yet with lower fluorescence intensity when compared to inner hair cells. We slightly rephrased the results section (page 6):

Upon dual-AAV injection into *Otof*^{-/-} cochleae, we found otoferlin immunofluorescence to be restricted to auditory HCs with stronger expression in IHCs and much weaker in sparsely transduced OHCs (Appendix Fig S3), resembling otoferlin expression in wild-type animals (Roux *et al*, 2006; Beurg *et al*, 2008).

Since we have not looked for otoferlin dual-AAV mediated vestibular hair cell transduction, we rephrased the respective sentence in the abstract:

Abstract:

In the cochlea, otoferlin was exclusively expressed in auditory hair cells.

Maximum confocal projections what does that mean?

We agree with the reviewer that we haven't exactly used the best choice of words here and replaced it with:

Maximum intensity projections of confocal optical sections,...

Arrows need to point at the cell, not the vicinity.

Done

no specific cell tropism.... Even within the organ of Corti I cannot see it in supporting cells, so the statement is unclear

We rephrased the sentence for clarification:

eGFP fluorescence was observed in IHCs, outer hair cells (OHCs), supporting cells, and spiral ganglion neurons (SGNs), indicating that the AAV2/6 has no specific IHC tropism and targets a variety of different cell types within the organ of Corti (Fig EV1A).

with the images now provided I am unable to verify the statement: was also found in other cell types, e.g. SGNs

As above, please see Fig EV1A,B, and especially B1 and B2 for eGFP expression in IHCs, supporting cells and spiral ganglion neurons. We rephrased the figure legend of Figure EV1 for further clarification.

for explaining the change in ABRs after injection:...might result from damage caused by the injection procedure and/or the pressure increase in the cochlea during the injection what is the difference between the two?

We agree that this sentence needs revision and more detailed information, as the pressure increase happens during the injection procedure. We rephrased the paragraph (page 15):

No difference between injected and non-injected contralateral wild-type ears was observed in ABR wave I amplitude and wave I-V latencies (Appendix Fig S4C,E). Even though we did not observe an apparent HC loss in our immunohistochemical analyses and click ABR thresholds were unaffected, a minor elevation of the 24kHz ABR threshold (Appendix Fig S4B) might point to a potential damage of HCs by the injection. The sensory cells of the inner ear are particularly sensitive to pressure and volume changes that might occur while injecting the virus solution into the cochlea (Yoshimura *et al*, 2018). In addition, we found a reduction in ABR wave II-V amplitudes in injected compared to non-injected ears (Appendix Fig S4A, D), the origin of which is unclear.

Instead of “can essentially restore auditory function”: How about can improve auditory function

We think that “improvement” is not the right term in this context as *Otof* knockout mice are profoundly deaf without treatment, similarly to the phenotype seen in most DFNB9 patients.

“Improvement” would imply that there is some residual hearing in these animals before treatment that could be augmented through otoferlin dual-AAV injection.

We rephrased the sentence to:

...and can at least partially restore auditory function.

References

- Beurg M, Safieddine S, Roux I, Bouleau Y, Petit C & Dulon D (2008) Calcium- and otoferlin-dependent exocytosis by immature outer hair cells. *J. Neurosci* **28**: 1798–1803
- Chatterjee M, Zion DJ, Deroche ML, Burianek BA, Limb CJ, Goren AP, Kulkarni AM & Christensen JA (2015) Voice emotion recognition by cochlear-implanted children and their normally-hearing peers. *Hearing Research* **322**: 151–162

- Fu QJ, Shannon RV & Wang X (1998) Effects of noise and spectral resolution on vowel and consonant recognition: acoustic and electric hearing. *J. Acoust. Soc. Am.* **104**: 3586–3596
- Fuchs PA & Lauer AM (2018) Efferent Inhibition of the Cochlea. *Cold Spring Harb Perspect Med*
- Huang L-C, Barclay M, Lee K, Peter S, Housley GD, Thorne PR & Montgomery JM (2012) Synaptic profiles during neurite extension, refinement and retraction in the developing cochlea. *Neural Dev* **7**: 1–17
- Landegger LD, Pan B, Askew C, Wassmer SJ, Gluck SD, Galvin A, Taylor R, Forge A, Stankovic KM, Holt JR & Vandenberghe LH (2017) A synthetic AAV vector enables safe and efficient gene transfer to the mammalian inner ear. *Nat. Biotechnol.* **35**: 280–284
- Luo X, Fu Q-J & Galvin JJ (2007) Cochlear Implants Special Issue Article: Vocal Emotion Recognition by Normal-Hearing Listeners and Cochlear Implant Users. *Trends in Amplification* **11**: 301–315
- McDermott HJ (2004) Music Perception with Cochlear Implants: A Review. *Trends in Amplification* **8**: 49–82
- Most T & Aviner C (2009) Auditory, Visual, and Auditory–Visual Perception of Emotions by Individuals With Cochlear Implants, Hearing Aids, and Normal Hearing. *J Deaf Stud Deaf Educ* **14**: 449–464
- Nelson PB, Jin S-H, Carney AE & Nelson DA (2003) Understanding speech in modulated interference: Cochlear implant users and normal-hearing listeners. *The Journal of the Acoustical Society of America* **113**: 961–968
- Oxenham AJ & Kreft HA (2014) Speech Perception in Tones and Noise via Cochlear Implants Reveals Influence of Spectral Resolution on Temporal Processing. *Trends in Hearing* **18** Available at: <http://tia.sagepub.com/cgi/doi/10.1177/2331216514553783> [Accessed July 13, 2015]
- Pangrsic T, Lasarow L, Reuter K, Takago H, Schwander M, Riedel D, Frank T, Tarantino LM, Bailey JS, Strenzke N, Brose N, Müller U, Reisinger E & Moser T (2010) Hearing requires otoferlin-dependent efficient replenishment of synaptic vesicles in hair cells. *Nat. Neurosci.* **13**: 869–876
- Paquette S, Ahmed GD, Goffi-Gomez MV, Hoshino ACH, Peretz I & Lehmann A (2018) Musical and vocal emotion perception for cochlear implants users. *Hear. Res.*
- Reisinger E, Bresee C, Neef J, Nair R, Reuter K, Bulankina A, Nouvian R, Koch M, Bückers J, Kastrup L, Roux I, Petit C, Hell SW, Brose N, Rhee J-S, Kügler S, Brigande JV & Moser T (2011) Probing the functional equivalence of otoferlin and synaptotagmin 1 in exocytosis. *J. Neurosci.* **31**: 4886–4895
- Roux I, Safieddine S, Nouvian R, Grati M, Simmler M-C, Bahloul A, Perfettini I, Le Gall M, Rostaing P, Hamard G, Triller A, Avan P, Moser T & Petit C (2006) Otoferlin, defective in a human deafness form, is essential for exocytosis at the auditory ribbon synapse. *Cell* **127**: 277–289
- Seibel NM, Eljouni J, Nalaskowski MM & Hampe W (2007) Nuclear localization of enhanced green fluorescent protein homomultimers. *Anal. Biochem.* **368**: 95–99
- Strenzke N, Chakrabarti R, Al-Moyed H, Müller A, Hoch G, Pangrsic T, Yamanbaeva G, Lenz C, Pan K-T, Auge E, Geiss-Friedlander R, Urlaub H, Brose N, Wichmann C & Reisinger E (2016) Hair cell synaptic dysfunction, auditory fatigue and thermal sensitivity in otoferlin Ile515Thr mutants. *EMBO J.* **35**: 2519–2535
- Yan Z, Zak R, Zhang Y & Engelhardt JF (2005) Inverted terminal repeat sequences are important for intermolecular recombination and circularization of adeno-associated virus genomes. *Journal of virology* **79**: 364
- Yoshimura H, Shibata SB, Ranum PT & Smith RJH (2018) Enhanced viral-mediated cochlear gene delivery in adult mice by combining canal fenestration with round window membrane inoculation. *Sci Rep* **8**: 2980

Thank you for the submission of your revised manuscript to EMBO Molecular Medicine. We have now received the enclosed reports from the referees that were asked to re-assess it. As you will see the reviewers are now globally supportive and I am pleased to inform you that we will be able to accept your manuscript pending following final editorial amendments.

***** Reviewer's comments *****

Referee #1 (Remarks for Author):

This revised manuscript is much improved. The authors have included supplemental data and made substantial changes to the manuscript in response to the reviewer's comments. In particular, the revised manuscript now includes sequencing results of additional PCR amplicons expressed in Oto^{-/-} mice as well as western blots (as requested) confirming expression of a shorter fragment which may compete with the full length protein. The authors appropriately discuss these new findings. Edits to the manuscript and supplemental experimental work are now included that clarify several points brought up by the reviewers. Overall, this manuscript is significantly improved. The results, describe here, will be of interest to investigators who are exploring novel gene therapy approaches targeting large genes to inner ear hair cells. The reviewer recommends the manuscript for publication.

Referee #2 (Comments on Novelty/Model System for Author):

I read the revised manuscript and the point by point response. My main concern was the lack of Otof KO transfected with AAV-eGFP as control for exocytosis and ABR. Now, the authors clearly mention that such a control has been already done in a previous study (Reisinger et al., 2011) and give detailed explanation. In addition, they respond to all the comment, in a very satisfy manner.

Referee #2 (Remarks for Author):

I read the revised manuscript and the point by point response. My main concern was the lack of Otof KO transfected with AAV-eGFP as control for exocytosis and ABRs. Now, the authors clearly mention that such a control has been already done in a previous study (Reisinger et al., 2011) and give detailed explanations. In addition, they responded positively to all the comments. From my side, the paper does not require additional changes and can be publish in the present form.

Referee #3 (Remarks for Author):

The authors performed a thorough and adequate revision and responded elegantly and positively to the comments of all 3 reviewers

Corresponding Author Name: Reisinger, Ellen and Kügler, Sebastian

Manuscript Number: 2018-09396